# Muscle calcium stress cleaves junctophilin1, unleashing a gene regulatory program predicted to correct glucose dysregulation

Eshwar R Tammineni[1], Lourdes Figueroa[1], Carlo Manno[1], Disha Varma[2], Natalia Kraeva[3], Carlos A Ibarra[3], Amira Klip[4], Sheila Riazi[3], Eduardo Rios[1]*

[1]Department of Physiology and Biophysics, Rush University, Chicago, United States; [2]Department of Internal Medicine, Division of Nephrology, Rush University, Chicago, United States; [3]Department of Anesthesia & Pain Management, University of Toronto, Toronto, Canada; [4]Cell Biology Program, The Hospital for Sick Children, Toronto, Canada

*For correspondence:
erios@rush.edu

Competing interest: The authors declare that no competing interests exist.

**Abstract** Calcium ion movements between cellular stores and the cytosol govern muscle contraction, the most energy-consuming function in mammals, which confers skeletal myofibers a pivotal role in glycemia regulation. Chronic myoplasmic calcium elevation ("calcium stress"), found in malignant hyperthermia-susceptible (MHS) patients and multiple myopathies, has been suggested to underlie the progression from hyperglycemia to insulin resistance. What drives such progression remains elusive. We find that muscle cells derived from MHS patients have increased content of an activated fragment of GSK3β — a specialized kinase that inhibits glycogen synthase, impairing glucose utilization and delineating a path to hyperglycemia. We also find decreased content of junctophilin1, an essential structural protein that colocalizes in the couplon with the voltage-sensing $Ca_V1.1$, the calcium channel RyR1 and calpain1, accompanied by an increase in a 44 kDa junctophilin1 fragment (JPh44) that moves into nuclei. We trace these changes to activated proteolysis by calpain1, secondary to increased myoplasmic calcium. We demonstrate that a JPh44-like construct induces transcriptional changes predictive of increased glucose utilization in myoblasts, including less transcription and translation of GSK3β and decreased transcription of proteins that reduce utilization of glucose. These effects reveal a stress-adaptive response, mediated by the novel regulator of transcription JPh44.

## Editor's evaluation

This is an important manuscript that provides evidence in favor of a possible mechanism that explains the lack of glucose utilization in skeletal muscle in pathological conditions (malignant hyperthermia) leading to hyperglycemia. It also describes an interesting compensatory mechanism via junctophilin 1 cleavage and gene expression.

## Introduction

In the excitation-contraction (EC) coupling process of striated muscles, action potential depolarization of the plasma or transverse (T) tubule membranes command the transient release of $Ca^{2+}$ into the myoplasm, enabling muscle contraction. The crucial device in this process is the couplon, a physical continuum of proteins that includes dihydropyridine receptor (DHPR, $Ca_V1.1$), ryanodine receptor 1

(RyR1), FKBP12, junctophilin1 (JPh1), Stac3, junctin, triadin, and calsequestrin, among other components (*Stern et al., 1997*; *Franzini-Armstrong et al., 1999*).

Only 5 proteins are necessary to assemble a functional, skeletal muscle-type EC coupling system in expression models (*Perni et al., 2017*; *Wu et al., 2018*). One of them is junctophilin, JPh (*Takeshima et al., 2000*), with skeletal and cardiac isoforms (1 and 2) deemed essential for creating and maintaining the junctional structure (T-SR junctions, dyads or triads) characteristic of striated muscle (*Komazaki et al., 2002*; *Rossi et al., 2019*).

Susceptibility to Malignant Hyperthermia (MHS *Litman et al., 2018*) is a condition paradigmatic of gain-of-function couplonopathies (*Ríos et al., 2015*; *Dirksen and Avila, 2004*). The primary defect in MHS is an alteration of RyR1 or another couplon protein, which leads to an increase in resting or activated openness of the channel. The increased 'leak' causes a reduction in free $Ca^{2+}$ concentration and total $Ca^{2+}$ content inside the SR, in turn leading to an increase in resting free $Ca^{2+}$ concentration, $[Ca^{2+}]_{cyto}$ (*Lopez et al., 1986*; *Lopez et al., 1992*), secondary to changes in the plasma membrane in response to SR depletion (*Eltit et al., 2004*; *Eltit et al., 2013*; *Manno et al., 2013*; *Ríos, 2010*). Duchenne Muscular Dystrophy (DMD) is another example of a condition that raises resting $[Ca^{2+}]_{cyto}$, albeit by different mechanisms (*Edwards et al., 2010*; *Boittin et al., 2006*; *Bellinger et al., 2009*).

MHS patients show other abnormalities, probably derived in many cases from the chronically elevated $[Ca^{2+}]_{cyto}$. They have altered musculoskeletal function (with pain, cramps, stiffness, muscle fatigue and bone deformity as common manifestations *Britt, 1991*; *Figueroa et al., 2019*), as well as systemic dysfunction. Notably, and possibly as a consequence of reduced uptake and processing of glucose by muscle (*Tammineni et al., 2020*), the frequency of hyperglycemia (*Altamirano et al., 2019*) and diabetes (*Tammineni et al., 2020*) is more than twofold greater in MHS patients than in the general age-matched population.

In 2013, Graham Lamb and coworkers reported the cleavage of JPh1 and 2 in skeletal muscle and JPh2 in cardiac muscle in the presence of high $[Ca^{2+}]_{cyto}$, and associated it to the activation of calpain 1 (*Murphy et al., 2013*). In cardiac muscle, the junctophilin isoform JPh2 is known to have a function similar to that of JPh1 in skeletal muscle — maintain structural integrity of the T-SR junction or dyad (*Takeshima et al., 2015*; *Lehnart and Wehrens, 2022*; *Perni, 2022*). Strikingly, fragments of JPh2 have been shown to additionally work in regulation of gene expression within nuclei (*Guo et al., 2018*; *Lahiri et al., 2020*).

We now find that the content of full-size JPh1 is reduced in skeletal muscle of MHS patients, while a C-terminal, 44 kDa JPh1 fragment, which we refer to as JPh44, moves away from the triadic location of the full-size protein and relocates to the nucleus. The observations indicate that the increased cleavage of JPh1 in MHS patients is due to activation of calpain1 by their higher $[Ca^{2+}]_{cyto}$.

The proteolysis of JPh1, with possible de-stabilization of T-SR junctions (*Komazaki et al., 2002*; *Ito et al., 2001*), and the entry of its fragments into nuclei could contribute to the disease phenotype, as proposed for fragments of JPh2 produced in cardiac muscle of failing hearts (*Guo et al., 2018*; *Lahiri et al., 2020*). On the other hand, LS Song and colleagues (*Guo et al., 2018*; *Wang et al., 2022*) provided evidence that the intra-nuclear actions of JPh2 fragments are 'stress-adaptive', partially offsetting the negative consequences of activation of proteolysis.

Given these precedents and the presence in the sequence of JPh44 of segments capable of interaction with DNA, here we tested the hypothesis that JPh44 exerts transcriptional control compensatory for the deleterious effects of elevated $[Ca^{2+}]_{cyto}$. The study demonstrates beneficial effects of the fragment on the transcription and translation of enzymes active in glucose utilization by skeletal muscle.

## Results

### JPh1 is cleaved in MH-susceptible patients

A large increase in $[Ca^{2+}]_{cyto}$ has been found in muscle fibers from MHS individuals (*Lopez et al., 1992*), myotubes derived from MHS patient biopsies (*Figueroa et al., 2019*) and in MHS animal models (*Lopez et al., 1986*). MHS animals also showed increased calpain activity (*Michelucci et al., 2017*). Given that junctophilins are targets of $Ca^{2+}$-activated calpain proteases (*Guo et al., 2018*; *Lahiri et al., 2020*), we hypothesized that excess $[Ca^{2+}]_{cyto}$, by promotion of calpain activity, enhances JPh1 cleavage in MHS individuals. As a test, JPh1 was quantified by Western blotting (WB) on the same 25-lane gel,

in total protein extracts from biopsied muscle of 13 MHN (normal) and 12 MHS subjects. The immuno-blot, stained using a polyclonal antibody raised against a human JPh1 immunogen (residues 387–512), referred to here as 'A', is shown in *Figure 1A*. A visible reduction in the MHS of a~72 kDa stained band revealed an almost twofold, statistically significant difference in content (*Figure 1C*). Because the 72–75 kDa migration weight is consistent with the full JPh1 sequence, we refer to this band as 'JPh1'. That the band is in some columns a doublet, and that it migrates slightly faster than the corresponding band in mouse muscle gels (*Figure 1—figure supplement 1*) indicates that proteolytic degradation may take place in the human biopsies and this band may not correspond to the full sequence of JPh1.

A blot from a different gel, stained with anti-JPh1 antibody 'B' (raised against a more distal mouse JPh1 immunogen, 509–622) showed a greater content of a band with an effective migration size of ~44 kDa, referred to as JPh44 (*Figure 1B and D*, p=0.06). Blots in *Figure 1—figure supplement 2* demonstrate that ab A reacts with both JPh1 and JPh44, while ab B fails to react with the 72 kDa protein, therefore constituting a useful tool for detection of JPh44 by immunofluorescence. The 72 kDa protein is overwhelmingly more abundant than its fragments, including JPh44 in nearly every condition (as shown in *Figure 1*, *Figure 1—figure supplement 2* and multiple documents later). Accordingly, the fluorescence of ab A can be used as a monitor of JPh1, with minimal error. Herein and unless noted differently, the labels 'JPh1' and 'JPh44' tag images of fluorescence of ab A and B, respectively.

While these results suggest that a greater cleavage of JPh1 leads to a reduction in its content, with consequent increase in the 44 kDa fragment, there is a poor negative correlation in individual subjects between the two changes (*Figure 1E*). A tentative explanation is that the cleavage process, which may operate at multiple sites and sequentially on cleaved products, results in faster degradation of the full protein, blurring any detailed correspondence between its disappearance and the increase in content of some fragments.

## Fragmentation of glycogen synthase kinase 3 (GSK3β) is increased in MHS patients

Phosphorylase kinase (PhK) reciprocally activates glycogen phosphorylase and inhibits glycogen synthase (GS). In previous work we demonstrated that this enzyme is activated in the muscle of MHS patients (*Tammineni et al., 2020*). Its effects are consistent with an observed shift of the glycogen ←→glucose balance towards glycogenolysis (*Tammineni et al., 2020*), which is presumed responsible for the hyperglycemia and diabetes that develops in many of these patients (*Tammineni et al., 2020*; *Altamirano et al., 2019*). Here, we compared the contents of GSK3β, a serine/threonine protein kinase that directly inhibits GS by phosphorylation. The kinase activity of GSK3β is controlled by its phosphorylation at Ser 9 and also promoted by truncation of the original ~47 kDa molecule to a form of ~40 kDa (*Jin et al., 2015*; *Goñi-Oliver et al., 2007*; *Ma et al., 2012*).

WB of protein extracts (*Figure 1F*) from muscle biopsies of the same patients of *Figure 1A*, reveal both forms of GSK3β. The signal in the 47 kDa band of full-size GSK3β was reduced in MHS, while that of the activated 40 kDa fragment was increased. The changes in the two proteins were highly negatively correlated (*Figure 1I*); consequently, the ratio of signals (40 kDa/47 kDa) was more than 3-fold greater in the MHS group, with high statistical significance. Regardless of diagnosis (MHS or MHN), there was a significant positive correlation between content of the activated GSK3β form and serum glucose (*Figure 1—figure supplement 3*). These results are consistent with activation of the GS kinase by calpain proteolysis, which contributes to the shift in glycogen ←→glucose balance in favor of glycogenolysis, adding to the previously reported effect of inhibitory phosphorylation of GS (*Tammineni et al., 2020*).

## Localization of the 44 kDa fragment of JPh1in skeletal muscle

In cardiac muscle, the full-length JPh2 protein is located at terminal cisternae of the SR, while cleaved JPh2 fragments can be found inside nuclei (*Guo et al., 2018*; *Lahiri et al., 2020*). Here we defined the location of JPh1 and JPh44 by immunostaining combined with 3D high-resolution imaging (a procedure used for every fluorescence image shown in this study, consisting in acquisition of vertical 'z-stacks' of confocal *x-y* images, followed by a correction algorithm described in Materials and methods). Moderately stretched thin myofiber bundles were fixed in PFA and stained with antibodies A or B. As shown in *Figure 2* (panels A), the ab A signal was largely located at T-SR junctions, where

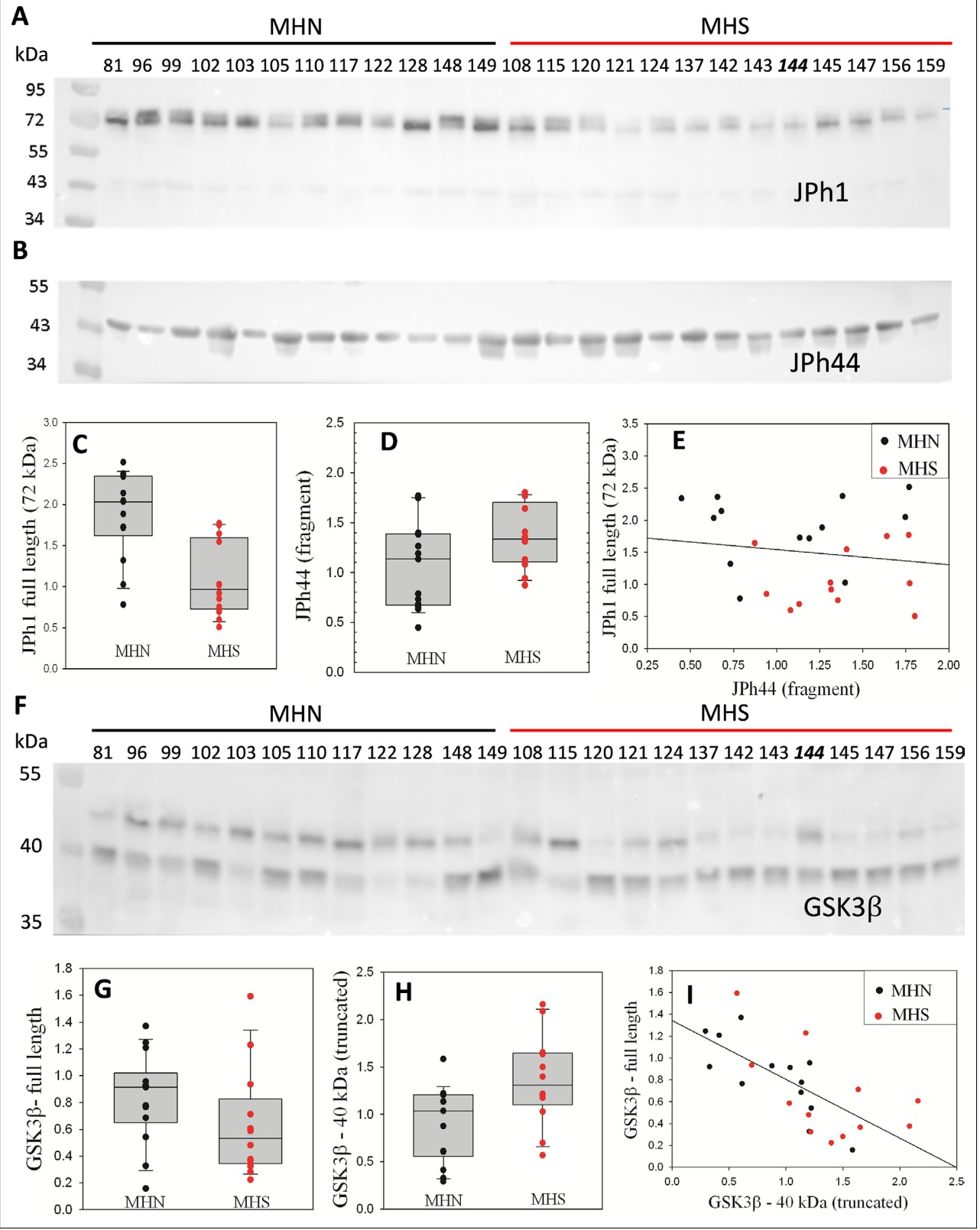

**Figure 1.** Fragmentation of proteins in MHS human muscle. (**A**) Western blot (WB) of whole protein fraction from individual subjects' biopsies, stained with antibody A. Subjects were diagnosed as MHN or MHS as indicated. Numbers identify individuals. (Sample #144, an MHN, is misplaced). (**B**) WB of a 2nd gel, using antibody B, which specifically stains JPh44 (see *Figure 1—figure supplement 2*). (**C, D**) Plots of signal in the 13 MHN and 12 MHS bands of panels **A** and **B**, respectively. p of no difference is <0.001 and 0.06, respectively for JPh1 and JPh44. (**E**) JPh1 vs JPh44 WB signals

*Figure 1 continued on next page*

*Figure 1 continued*

for all patients. The correlation is poor (R=–0.16). (**F**) WB of whole protein fraction from the same biopsies, stained for GSK3β. (**G, H**) Box plots of the signal in the ~47 and~40 kDa bands for MHN and MHS subjects. The difference is highly significant for the light band signal (p=0.015) but not for the 47 kDa band (P=0.14). (**I**) 40 kDa vs. 47 kDa signals for all patients. The correlation is highly significant (R=–0.69, p<0.001). *Data trace*. Raw data and statistical calculations in *GSK3β graphs and statistics.JNB*, sections 1 and 2 (all data in Harvard Dataverse described in Materials and methods). See also *Figure 1—figure supplements 1–3*.

The online version of this article includes the following source data and figure supplement(s) for figure 1:

**Source data 1.** JPh1 raw blot shown in *Figure 1A*.

**Source data 2.** Gel for JPh1 blot in *Figure 1A*.

**Source data 3.** JPh44 raw blot shown in *Figure 1B*.

**Source data 4.** Gel for JPh44 blot of *Figure 1B*.

**Source data 5.** Raw blot in *Figure 1F*.

**Source data 6.** Gel for blot of *Figure 1F*.

**Figure supplement 1.** Staining of muscle extracts from mice and a human subject, by antibody A, raised against a human JPh1 immunogen (residues 387–512).

**Figure supplement 1—source data 1.** Raw blot of JPh1 (abA).

**Figure supplement 1—source data 2.** Originating gel (right half) for raw blot of JPh1.

**Figure supplement 2.** Specificity of junctophilin antibodies.

**Figure supplement 2—source data 1.** Different amounts of human muscle total extract, blot stained with Junctophilin abA as presented in *Figure 1—figure supplement 2*.

**Figure supplement 2—source data 2.** Different amounts of human muscle total extract, blot with Junctophilin abB as presented in *Figure 1—figure supplement 2*.

**Figure supplement 2—source data 3.** Originating gel for blots in *Figure 1—figure supplement 2*.

**Figure supplement 3.** Plots of Fasting Blood Glucose (FBS) measured in the 25 patients with analysis of GSK3β analysis reported in *Figure 1*, vs. the normalized content of truncated (activated) GSK3β (panel **A**), or the truncated/full-size ratio of contents (panel **B**).

it was highly colocalized with RyR1. In contrast, ab B (JPh44-specific) detected discrete particles at variable locations within the sarcomeric I band, as well as nuclei (panel 2Bb), and accordingly failed to colocalize with RyR1 (panels 2 Cc, Cd).

Colocalization between the junctophilin forms and RyR1 was quantified by four measures, with average results listed in *Table 1*. First: R, Pearson's correlation coefficient of immunofluorescence intensities, calculated pixel by pixel after subtraction of a small background signal. This index had high value for the ab A fluorescence, (indicating a much greater quantity of full-size JPh1 than JPh44) and low for ab B (JPh44), with high statistical significance of the difference. Second: the Intensity Correlation Quotient (ICQ), emergent from an approach by *Li et al., 2004* illustrated in panel 2Cb, whereby the intensity of fluorescence of ab A (JPh1) is plotted, pixel by pixel, against the covariance of ab A and RyR1 intensities (Materials and methods). The comma-shaped cloud with negative curvature, shown in 2Cb, is a characteristic of high colocalization, contrasting with the funnel-shaped cloud found for ab B vs RyR1 (panel 2 Cd), which includes pixels with negative covariance, reflecting mutual exclusion rather than colocalization. The ICQ, which varies between –0.5 (reflecting exclusion) and 0.5 (perfect colocalization), was close to 0.5 for ab A, while that of ab B was smaller but still positive, a difference with high statistical significance (*Table 1*; here and elsewhere significance was established through biological replications).

A third approach, introduced by *van Steensel et al., 1996* and illustrated in *Figure 2* panels Ca, c, plots the correlation coefficient of the two signals, averaged over all pixels, vs. a variable shift of one of the images in one direction (in the example, the shift is in the x direction, parallel to the fiber axis). The curve generated is then fitted by a Gaussian. The x-axis location of the apex of the Gaussian (or that of the actual correlation, when the fit is poor), named 'VS shift', provides a rough measure of the average separation of the two fluorescent species in the x direction. In the example, the distance was 21 nm for JPh1 (antibody A, panel 2Ca) and 154 nm for JPh44 (2 Cc), a significant difference (*Table 1*). Fourth: the FWHM of the fitted Gaussian is a second measure of dispersion or de-localization, also greater for the JPh fragment. Further use of these techniques will show that the calculation of four

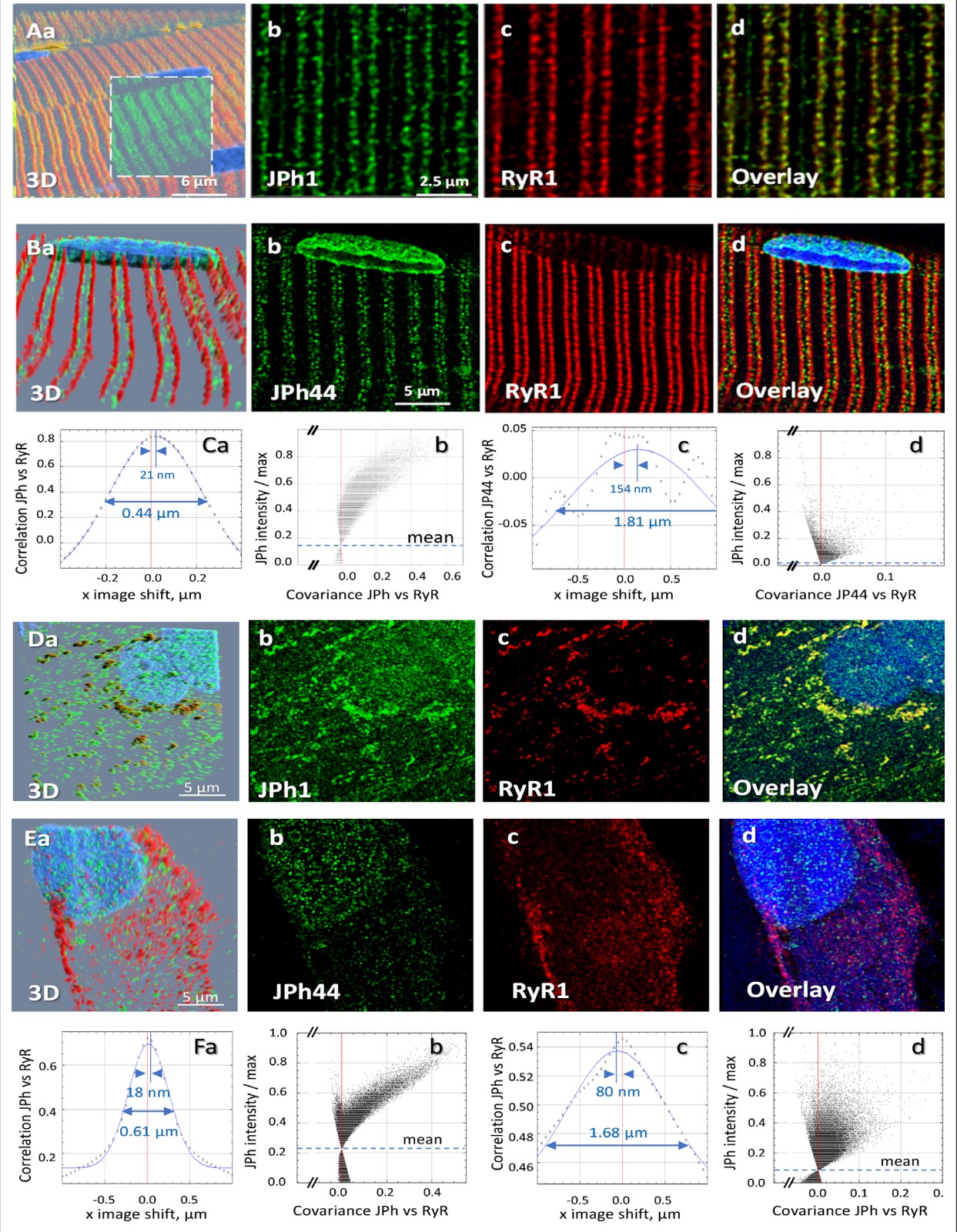

**Figure 2.** Location of two forms of junctophilin in human skeletal myofibers and myotubes. (**A, B**) Images respectively stained with antibodies A and B, in cells co-stained for RyR1 (Overlay panels add nuclear staining). JPh1 (ab A) and Ryr1 (panels **Ab**, **c**) are highly colocalized at T-SR junctions. JPh44 (ab B) moves away into the I band (panels **Bb**, **d**). Colocalization between JPh1 and RyR1 is high, as demonstrated by the Van Steensel's and Li's analyses (panels **Ca**, **b**), leading to 4 measures (R, VS shift, FWHM and ICQ) listed in *Table 1*. As indicated in Ca, VS shift is 21 nm and FWHM is 0.44 μm.

*Figure 2 continued on next page*

*Figure 2 continued*

Colocalization of JPh44 and RyR2 is lost, as indicated by a VS shift of 154 nm, a FWHM of 1.8 μm defined on a Gaussian that fits poorly an oscillating VS plot and a Li plot characteristic of no colocalization and exclusion (compare panels **Cb** and **Cd**). (**D, E**) Primary myotubes derived from patients' muscle, stained respectively with abs A and B, and co-stained for RyR1. Note in panels **D**, overlap of JPh1 (ab Al) with cytosolic RyR1. In (**E**) JPh44 (ab B) is largely intra-nuclear, with no colocalization with RyR1. Visual impression is supported by colocalization analyses (panels **F**, replication results in *Table 1*). *Data trace*. Average colocalization measures are listed in *Table 2*. Experiment identifiers: panel **A**, 102919La Series 010 (ID 167, HH); panel **B**, 102519 L Series 018; panels **D, G**, 072420Lb Series 002; panel **E**, 030120Lb Series 001. Data in *Summary ratios.xlsx*.

The online version of this article includes the following video and figure supplement(s) for figure 2:

**Figure supplement 1.** IF images from the human fiber bundle illustrated in *Figure 2*, stained with the anti-JPh1 ab A, and RyR1 and nuclear markers.

**Figure 2—video 1.** 3D rendering of MHN muscle stained with Ab B and RyR1.

https://elifesciences.org/articles/78874/figures#fig2video1

**Figure 2—video 2.** 3D rendering of MHS muscle stained with Ab B and RyR1.

https://elifesciences.org/articles/78874/figures#fig2video2

**Figure 2—video 3.** 3D rendering of MHN muscle stained with Ab A and RyR1.

https://elifesciences.org/articles/78874/figures#fig2video3

**Figure 2—video 4.** 3D rendering of MHS muscle stained with Ab A and RyR1.

https://elifesciences.org/articles/78874/figures#fig2video4

**Figure 2—video 5.** 3D rendering of MHS muscle stained with Ab B and RyR1.

https://elifesciences.org/articles/78874/figures#fig2video5

**Table 1.** Colocalization of JPh1 (as reported by the ab A signal) and its fragment (ab B) with RyR1 in muscle or muscle-derived myotubes.

Col. 1: Pearson's pixel-by-pixel correlation coefficient. 2: Van Steensel's shift in the x direction (parallel to fiber axis, arbitrary direction for myotubes). 3: Van Steensel's FWHM of Gaussian fit. 4: Li's Intensity Correlation Quotient. Definitions in Materials and methods. p, probability of no difference between values for JPh44 and JPh1. There was no statistically significant difference between colocalization measures of the same protein in myotubes vs myofibers (p's not shown). *Data trace*: raw data and statistics. In *JPh/manuscript /ColocalizJP44andRyR.JNB*.

| | | | | 1 | | 2 | 3 |
|---|---|---|---|---|---|---|---|
| | | N subjects | N, fibers or myotubes | R | ICQ | VS shift, nm | FWHM, μm |
| JPh44 in myofibers | average | 3 | 10 | 0.31 | 0.22 | 66 | 0.56 |
| | median | | | 0.26 | 0.23 | 54 | 0.50 |
| | S.E.M. | | | 0.04 | 0.02 | 15 | 0.19 |
| JPh1 in myofibers | average | 3 | 8 | 0.88 | 0.4 | 16 | 0.42 |
| | median | | | 0.9 | 0.42 | 12 | 0.40 |
| | S.E.M. | | | 0.03 | 0.02 | 3 | 0.07 |
| p | | | | <0.001 | <0.001 | <0.001 | 0.06 |
| JPh44 in myotubes | average | 2 | 9 | 0.37 | 0.22 | 168 | 1.74 |
| | median | | | 0.36 | 0.22 | 164 | 1.69 |
| | S.E.M. | | | 0.05 | 0.03 | 36 | 0.17 |
| JPh1 in myotubes | average | 3 | 29 | 0.73 | 0.32 | 38 | 0.55 |
| | median | | | 0.77 | 0.32 | 38 | 0.50 |
| | S.E.M. | | | 0.02 | 0.01 | 4 | 0.03 |
| p | | | | <0.001 | 0.003 | <0.001 | <0.001 |

different measures provides a multidimensional view of colocalization — or its absence — revealing differences between proteins and treatments not reflected in the usual correlation analysis.

Images of patient-derived primary myotubes, stained with antibodies A and B and co-stained for RyR1, are illustrated in *Figure 2D and E*. Here too, the differences were clear. Ab A staining in cytosol was highly colocalized with RyR1 (2D, 2 F, quantification in *Table 1*), in clusters that presumably correspond to developing junctions. In agreement with the observation in myofibers, ab B did not colocalize with RyR1 and was found mostly within nuclei, in fine granular form (panel 2Eb). Unlike myofibers, myotubes had some intra-nuclear staining with ab A, which adopted a fine granular pattern similar to that of ab B. Observations presented later assign this ab A signal to JPh44.

## The 44-kDa fragment internalized in nuclei includes the C terminus of JPh1

In earlier studies of JPh2, both N-terminal (*Guo et al., 2018*) and C-terminal fragments (*Lahiri et al., 2020*) were shown to translocate to nuclei, in conditions of heart stress. To identify the JPh1 segment cleaved as JPh44 and follow its movements, a fusion of JPh1 with GFP (at the N terminus) and the FLAG tag (at the C terminus) was expressed in myotubes derived from patients' muscle or a C2C12 line, and in mouse adult myofibers. Panels A and B in *Figure 3*, of myotubes expressing GFP-JPh1-FLAG, show GFP fluorescence (green) largely in the cytosol, in the form of small clusters or puncta. Instead, the C-terminal FLAG (red) appeared in both nuclear and extra-nuclear regions — corresponding to the cytosol and other organelles. The intensity of the intranuclear FLAG signal was highly variable in individual nuclei (*Figure 3—figure supplement 1*). Within the cytosol, the FLAG tag was distributed in two forms: punctate, colocalized with (N-terminal) GFP fluorescence, as well as a finely particulate disperse form, away from GFP, obviously corresponding to a fragment cleaved at a position distal to the N terminus.

The asymmetry was quantified by the ratio of signal densities, extranuclear (loosely called 'cytosolic')/nuclear, which in C2C12 myotubes was approximately threefold greater for GFP, a highly significant difference (*Table 2*). When the plasmid was expressed in myotubes derived from human muscle, a difference greater than twofold, again in favor of GFP, was found (panels 3B and *Table 2*). A clear nuclear internalization of FLAG, with exclusion of GFP, was also found in adult mouse muscle expressing the plasmid (*Figure 3C*). In this case, the ratio (cytosolic/nuclear) of signal densities was nearly 1000-fold greater for the N-terminal GFP than for the C-terminal FLAG (*Table 2*).

An additional observation (illustrated with panels 3Ca, d, e, and repeated in replications) was that in myofibers of adult mice the C-terminal fragment accumulated in perinuclear regions, in addition to entering nuclei. This was not the case for the native JPh44 in myofibers (*Figure 2*), which suggests that the perinuclear buildup be a consequence of the steep gradient generated by the transient increase of concentration of the fragment, combined with a diffusion barrier at the nuclear membrane. The accumulation might also reflect a sequence difference between the exogenous piece and JPh44. This explanation is refuted however, by the absence of perinuclear FLAG accumulation in myotubes expressing the construct (*Figure 3Ad*, Bd), which instead suggests a slower production of the protein or a lower barrier to nuclear entry in developing cells.

In the transfected mouse myofibers we also found the N-terminal GFP strictly colocalized with endogenous RyR1 at triads, reproducing the high colocalization of endogenous JPh1 and RyR1 found in human myofibers (illustrated with *Figure 2* and *Figure 2—figure supplement 1*). These observations allow three conclusions: (1) in both developing and adult tissue, the full-size protein is attached to junctions between plasma or T membrane and SR. (2) The C terminus is part of the cleaved fragment that migrates into the I band and enters nuclei. (3) The N terminus of JPh1 stays with the plasma or T membrane, whether as part of the full protein or after cleavage.

## Elevated cytosolic calcium concentration mediates the increase in JPh1 cleavage

Resting $[Ca^{2+}]_{cyto}$ in myofibers of MHN and MHS human subjects was reported as 112 nM and 485 nM, respectively (*Lopez et al., 1992*). (*Murphy et al., 2013*) reported some cleavage of JPh1 at $[Ca^{2+}]_{cyto}$ of 500 nm. Hypothesizing that the increased cleavage of JPh1 in MHS subjects is associated with increased $[Ca^{2+}]_{cyto}$, we quantified the effect of this level of cytosolic $[Ca^{2+}]$ on the fragmentation of JPh1. Biopsied myofiber bundles from three subjects, pinned in chambers and exposed to saponin

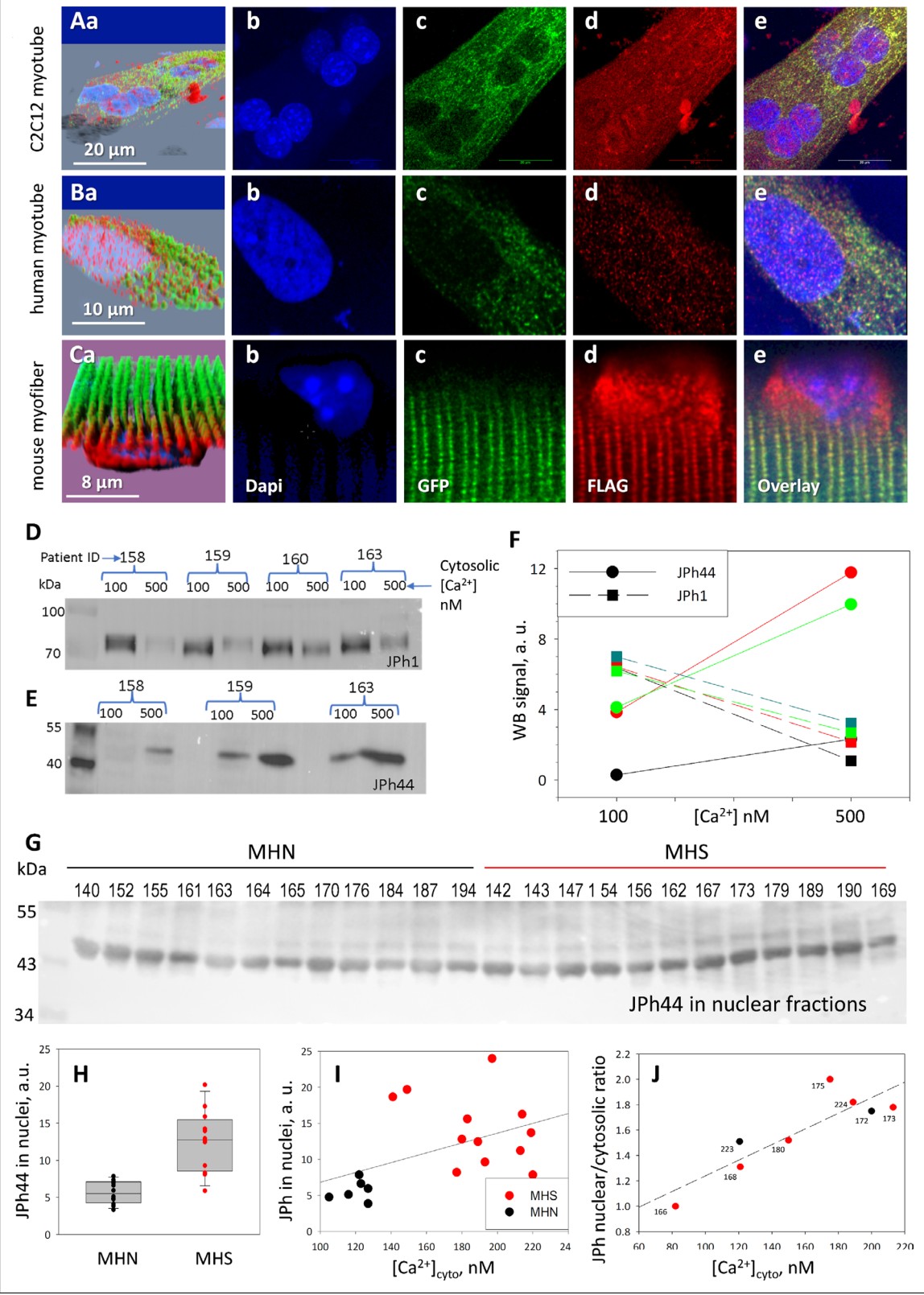

**Figure 3.** Distribution of a dually tagged JPh1 and its fragments. (**A-C**) Confocal images of cells expressing the (N)GFP-JPh1-FLAG(C) construct. In myotubes (**A, B**), GFP is exclusively in the cytosol, while FLAG, marking the full-size protein and its C-terminal fragment(s), red, distributes widely, including inside nuclei. The different distribution is also evident in adult myofibers (panels **C**), with the additional observation of accumulation of the N-terminal fragment in perinuclear regions (Cd). Panels **Aa**, **Ba** and **Ca** -- 3D views of the image stacks -- show the intranuclear location of FLAG. Full

*Figure 3 continued on next page*

*Figure 3 continued*

stacks are viewable as Supp. videos to *Figure 3*. The distribution differences are highly significant (details in *Table 3*). (**D-F**) Effect of extracellular $Ca^{2+}$ on JPh1 content of permeabilized muscle from four subjects. JPh1 (ab A) and the 44 kDa fragment (ab B) changed reciprocally in muscles exposed to 500 vs 100 nM $[Ca^{2+}]$. (**G**) JPh44 (stained by ab B) in whole muscle nuclear fraction extracted from 12 MHN and 12 MHS individuals. H, the average content is more than twofold greater in MHS (p<0.001). (**I**) JPh44 content in nuclei (from WB in panel **G**) vs. $[Ca^{2+}]_{cyto}$ in primary myotubes from the same muscle samples, showing a statistically significant positive correlation (R=0.48, p=0.04). (**J**) Ratio JPh content in nuclei / JPh in cytosol, in images of myotubes stained with ab A, vs. $[Ca^{2+}]_{cyto}$. The correlation is positive and statistically significant (R=0.88, p=0.004). *Data trace*: **A**: experimental record 091620 a Series 5 Lng; **B**: 091020 a Series 4 Lng, from patient MHN #179; **C**: 100520 a Series 2. **F**, **H–J** in *JPh vs Ca and GSK3b vs FSB.JNB*. **F** in Section 1, **I** in Section 2, **J** in Section 3.

The online version of this article includes the following video, source data, and figure supplement(s) for figure 3:

**Source data 1.** JPh1 raw blot in *Figure 3D*.

**Source data 2.** Originating gel for blot in *Figure 3D*.

**Source data 3.** Raw blot in *Figure 3E* (boxed region).

**Source data 4.** Gel for *Figure 3E*.

**Source data 5.** Raw blot for *Figure 3G*.

**Source data 6.** Originating gel for blot in *Figure 3G*.

**Figure supplement 1.** Muscle bundle from adult, presumably normal mouse expressing GFP-JPh1-FLAG.

**Figure 3—video 1.** The z-stack from which panels A in *Figure 3* were derived, as animated series.
https://elifesciences.org/articles/78874/figures#fig3video1

**Figure 3—video 2.** The z-stack from which panels B in *Figure 3* were derived, as animated series.
https://elifesciences.org/articles/78874/figures#fig3video2

**Figure 3—video 3.** The z-stack from which panels C in *Figure 3* were derived, as animated series.
https://elifesciences.org/articles/78874/figures#fig3video3

for permeabilization, were superfused with either 100 or 500 nM $[Ca^{2+}]$. After 10 min of exposure, the bundles were processed separately to extract protein (hereon 'whole-protein extracts'). Quantitative WB of the extract found increased JPh1 cleavage upon exposure to higher $[Ca^{2+}]$ (*Figure 3D–F*).

Together with an excess JPh44 in whole-protein extracts from MHS patients' muscle, we found a significantly higher JPh44 content in nuclear fractions from these biopsies (*Figure 3G and H*). The altered phenotype of MHS patients, including elevated $[Ca^{2+}]_{cyto}$, is reproduced to a large extent in myotubes derived from their muscle biopsies (*Figueroa et al., 2019*; *Figueroa et al., 2021*). Two remarkable observations affirm the relevance of the elevated cytosolic calcium in defining the altered

**Table 2.** Fate of doubly tagged protein GFP (N) – JPh1 – FLAG (C).

Distribution of GFP (N terminal) and FLAG (C-terminal) in different preparations, quantified by the ratio of densities in extranuclear ('cytosol') and nuclear areas. N, numbers of culture experiments, patients or mice. n, numbers of images. nn, numbers of nuclei included in calculations. p, probability of no difference between GFP and FLAG distributions in two-tailed t tests. To correct for data clustering (or pseudoreplication), average and dispersion parameters in mouse myofibers were derived by hierarchical analysis. *Data trace*: Raw data listed in depository under the following identifiers: C2C12: 091620 a. Series 2, 5, 7. Patient-derived myotubes: 091020 a. Series 1–4. Mice: 092420 a, 100520 a, 100520b. All series. Murine in *summary flag-gfp.JNB*. Patient in *ColocalizJP44and RyR.JNB* in JPh/Manuscript Data2.

| Cytosolic/ nuclear density | GFP | | | FLAG | | |
|---|---|---|---|---|---|---|
| | C2C12 | Patient-derived myotubes | Murine myofibers | C2C12 myotubes | Patient-derived myotubes | Murine myofibers |
| avg | 2.32 | 3.13 | 139 | 0.84 | 1.44 | 0.15 |
| median | 2.47 | 2.62 | 45.5 | 0.87 | 1.4 | 0.09 |
| sem | 0.47 | 0.59 | 38.5 | 0.12 | 0.26 | 0.03 |
| N, n, nn | 1, 3, 6 | 1, 4, 38 | 3, 16, 20 | 1, 3, 6 | 1, 4, 38 | 3, 16, 20 |
| P | | | | 0.039 | 0.028 | <0.001 |

phenotype. First, there was a positive correlation between the nuclear JPh44 content in the muscle of patients and $[Ca^{2+}]_{cyto}$ in their derived myotubes (*Figure 3I*). There was also a significant positive correlation between nuclear content of JPh in the myotubes (as quantified by the nuclear/cytosolic ratio of JPh densities) and their $[Ca^{2+}]_{cyto}$ (*Figure 3J*). Taken together, the observations indicate that cleavage of JPh1 and nuclear internalization of the 44 kDa fragment are higher in MHS muscle, driven by the higher $[Ca^{2+}]_{cyto}$ found in these patients.

## Calpain1 cleaves JPh1 to produce a 44 kDa C-terminal fragment

*Murphy et al., 2013* showed that activation of calpain 1, a heterodimer that incudes a main (~80 kDa) subunit, is associated with autocatalytic proteolysis and results in cleavage of junctophilins 1 and 2. Later studies confirmed that JPh2 in mammalian heart is cleaved by calpains (*Guo et al., 2018*; *Lahiri et al., 2020*), most effectively by calpain1 (*Yoshimura et al., 1983*), activated upon autolysis to a~76 kDa isoform (*Goll et al., 2003*; *Suzuki et al., 1981*; *Moldoveanu et al., 2002*). Because this activation is reported to occur in the 50–300 nM range of $[Ca^{2+}]_{cyto}$, we tested whether calpain1 autolysis occurs and is responsible for the increased cleavage of JPh1 in MHS.

The prediction tool GPS-CCD 1.0 (*Liu et al., 2011*) applied to JPh1 — the human sequence of which (*Yang et al., 2022*) is diagrammed in *Figure 4A* — revealed several conserved calpain1 cleavage sites in multiple JPh1 orthologs (*Figure 4B*). The one with highest score is at R240-S241, within the cytosolic domain, in-between MORN motifs 6 and 7 (*Figure 4A–C*). In all these orthologs, cleavage at the R-S site will generate a C-terminal fragment of between 45.92 and 46.28 kDa, consistent with the ~44 kDa migration size of JPh44 and its C-terminal location in the sequence of JPh1.

To test whether the greater cleavage of JPh1 in MHS is due to activation of calpain1, we analyzed by Western blotting the calpain1 content of muscle from the subjects with JPh content represented in *Figure 1*. The immunoblot revealed a double band, with a component at ~80 kDa and another at ~76 kDa, the sizes of the protease and its activated fragment (*Figure 4E*). We adapted a custom method to quantify immunoblots (*Tammineni et al., 2020*) to automatically compute the signal in closely placed double bands. Details of the procedure are in *Figure 4—figure supplement 1*. Comparisons of the signal in the two bands between MHS and MHN patients are represented in *Figure 4F*. The signal differences between MHS and MHN were not statistically significant for either calpain band. However, the paired ratios of band signals (76 kDa / 80 kDa, *Figure 4G*) were on average 74% higher in the MHS (p=0.029). This ratio, which can be taken as a measure of calpain activation, was negatively correlated with both the (72 kDa) JPh1 and GSK3β contents, and positively correlated with content of the 40 kDa GSK3β fragment, quantified in the same muscle samples (*Figure 4H–J*). The relative increase in the 76 kDa fragment was also accompanied by a higher content of smaller polypeptides detected by the calpain1 antibody (*Figure 4—figure supplement 2*).

To test the ability of calpain1 to cleave human JPh1, total protein extracts from patients' muscle were incubated with 0.5 units of calpain1 at different time intervals (*Figure 5A and B*). Incubation resulted in cleavage, as reflected in reduction in content of full-length JPh1 and increase of the JPh1 44 kDa fragment (still referred to as 'JPh44', even though it was here obtained by a different procedure). The changes were essentially completed in 5 min.

The effects of calpain 1 on the two forms were large and reciprocal, and increased with protease concentration (5 C, D).

Finally, calpain-dependent JPh1 fragmentation was effectively inhibited in the presence of the calpain inhibitor MDL 28170 (*Figure 5E and F*). While the observation is consistent with calpain1-specific cleavage of human JPh1, it does not exclude effects of calpain 2 or cathepsin B, as these are also targets of the inhibitor.

While calpain1 is freely diffusible in the presence of normal cytosolic concentrations of $Ca^{2+}$, it binds to cellular structures immediately upon elevation of $[Ca^{2+}]_{cyto}$, which implies that its proteolytic activity can only be exerted on substrates that are close-by at the time of activation (*Murphy et al., 2006b*; *Murphy, 2009*). Therefore, to target JPh1 effectively, calpain1 must be present near T-SR junctions. To define this location precisely, human myofibers and human-derived primary myotubes were stained for calpain1, junctophilin (using antibodies A or B), and RyR1 as junction marker, and 3D-imaged at high resolution. Images of endogenous calpain1 in myofibers show that it colocalized with JPh1 (*Figure 5G*, *Figure 5—figure supplement 1*) — confirming the presence of the protease near junctions — while JPh44 was found in the I band, not colocalized with calpain1 (5 H). The quantitative

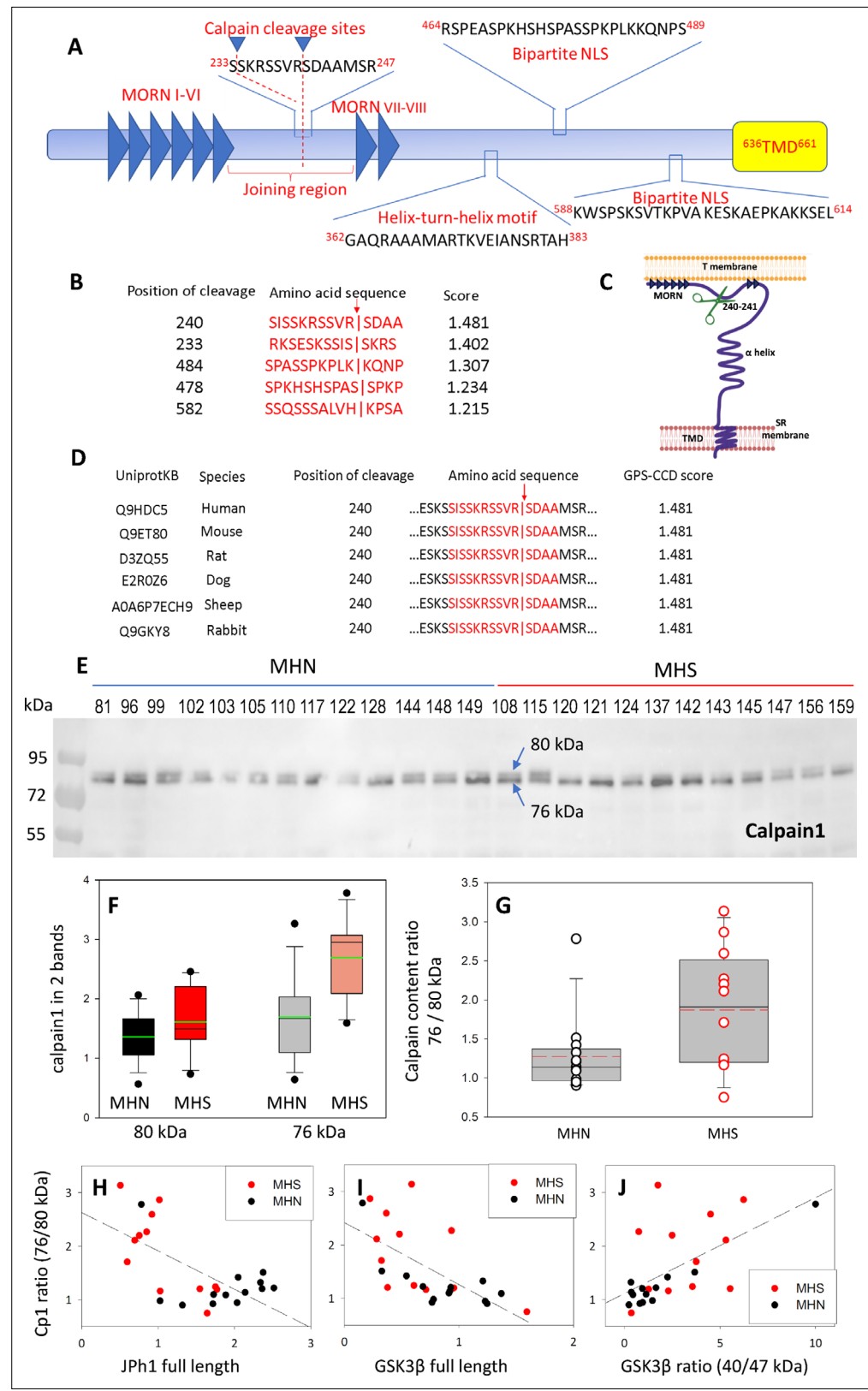

**Figure 4.** Activation and activity of calpain 1 in muscles of MHS patients. (**A**) JPh1 sequence indicating conserved stretches and the two calpain cleavage sites with highest priority score, located between MORN motifs VI and VII. (**B**) The 5 top-priority cleavage sites predicted by GPS-CCD 1.0 (*Li et al., 2004*). (**C**) JPh1 secondary structure, showing location of the highest priority cleavage site (R240-S241), helix-turn-helix DNA interaction site and TMD in

*Figure 4 continued on next page*

*Figure 4 continued*

SR junctional membrane. (**D**) Conservation of the preferred cleavage sites R240-S241 and S233-S234 in mammalian orthologs, which produce in every case a C-terminal fragment of ~44 kDa. (**E**) WB of whole-tissue protein fraction in biopsied muscle of 13 MHN and 12 MHS patients, showing a dual band at ~80 kDa (details in *Figure 4—figure supplement 1*). (**F**) Box plot of full-size and 76 kDa truncated forms, compared in the MHN and MHS groups. (**G**) Distribution of ratios of 76/80 kDa forms. Median ratio in MHS (1.97) is 74% greater than that in MHN (1.13) with p=0.029 of no difference. Automatic quantification of the WB is in *Figure 4—figure supplement 2*. (**H, I, J**) Correlation of ratios of 76/80 kDa junctophilin 1 with contents of full-length JPh1 and GSK3β, and ratio of activated to full-length forms of GSK3β, derived by WB of the same muscle fractions used for the blot in panel **E**. The correlation coefficients R and p of no correlation are: for **H** –0.65 and <0.001; for **I** –064 and <0.001; for **J** 0.61 and 0.001. *Data trace*: Panels **E-G** and **H-J** respectively in sections 6 and 5 of *Calpain files with correlations ER IDL. JNB*.

The online version of this article includes the following source data and figure supplement(s) for figure 4:

**Source data 1.** Raw blot for *Figure 4D* (boxed region).

**Source data 2.** Originating gel for blot in *Figure 4D*.

**Figure supplement 1.** A streamlined method to quantify double bands in western blots.

**Figure supplement 2.** Western blot of calpain1 in whole-tissue protein of human subjects.

**Figure supplement 2—source data 1.** Raw blot (boxed area in pdf document).

**Figure supplement 2—source data 2.** Originating gel.

---

measures described for colocalization of JPh and RyR1 were also indicative of colocalization of calpain1 with JPh1, but not with JPh44 (*Figure 5I–K* and *Table 3*).

## Effects of exogenous calpain on living cells

The effects were studied by heterologous expression of tagged calpain1 in patient-derived myotubes. The main effect of the overexpressed calpain1 was an increase in intranuclear localization of JPh (*Table 4*), illustrated in *Figure 6* with images of a human-derived culture transfected with FLAG-calpain1 and additional comparisons with control cultures in *Figure 6—figure supplement 1*. Panels 6A-D are 3D renderings of the *z*-stacks of confocal images. A and B show FLAG-calpain1 and junctophilin stained with ab A. Panel C renders RyR1 in the same cells. The images allow direct comparison, within the same field, of a myotube (Cell 1) that expressed strongly the FLAG-tagged calpain1, and one (Cell 2) that had no trace of the protein. Calpain1 (panel A) was widely distributed in the cytosol; nuclei can be recognized in A and C by the absence of calpain1 and RyR1. While the ab A signal (panel B) was widely distributed in Cell 2, in Cell 1 it resided mostly inside nuclei, presumably because it tagged largely JPh44, cleaved by the excess calpain. The intranuclear junctophilin in Cell 1 adopted a diffuse, fine-grained form, while the cytosolic junctophilin in Cell 2 often formed puncta or clusters. The increase in nuclear junctophilin in cells expressing calpain was large and highly significant (*Table 5*).

RyR1 is clustered in both cells (*Figure 6C and G*). The overlay (D) shows that the ab A signal colocalizes with RyR1 (red) in Cell 2, corresponding to the more abundant JPh1 full-size form, but not in calpain-expressing Cell 1, where the signal is largely inside nuclei, presumably marking the JPh44 fragment. As illustrated with *Figure 6—figure supplement 1*, colocalization of JPh and RyR1 was stronger in control cells (not transfected with FLAG-calpain1), with highly significant differences in all 4 measures (*Table 5*).

Panels 6E-G show fluorescence in one slice of the *z*-stack, with pairwise overlays in H-J. The calpain1-rich cytosol of Cell 1 shows JPh1 in fine granular form, without clusters (Panel H). The same region shows abundant clusters of RyR1 (6 G, I). Cell 2, which lacks exogenous calpain1, shows abundant JPh1 in clusters (F), in almost perfect colocalization with RyR1(panel J), which identifies these puncta as JPh1 in developing T-SR junctions.

In Cell 1, where calpain1 was expressed, the protease was also clustered (panel E), highly colocalized with RyR1, that is, located at T-SR junctions (panel 6 I, colocalization quantified in *Table 6*). As argued by *Murphy, 2009*, the location of calpain1 at calcium release sites is consistent with the idea that Ca²⁺ activation of calpain1 causes its binding to structures, an effect that places the protease at an ideal location for cleaving JPh1.

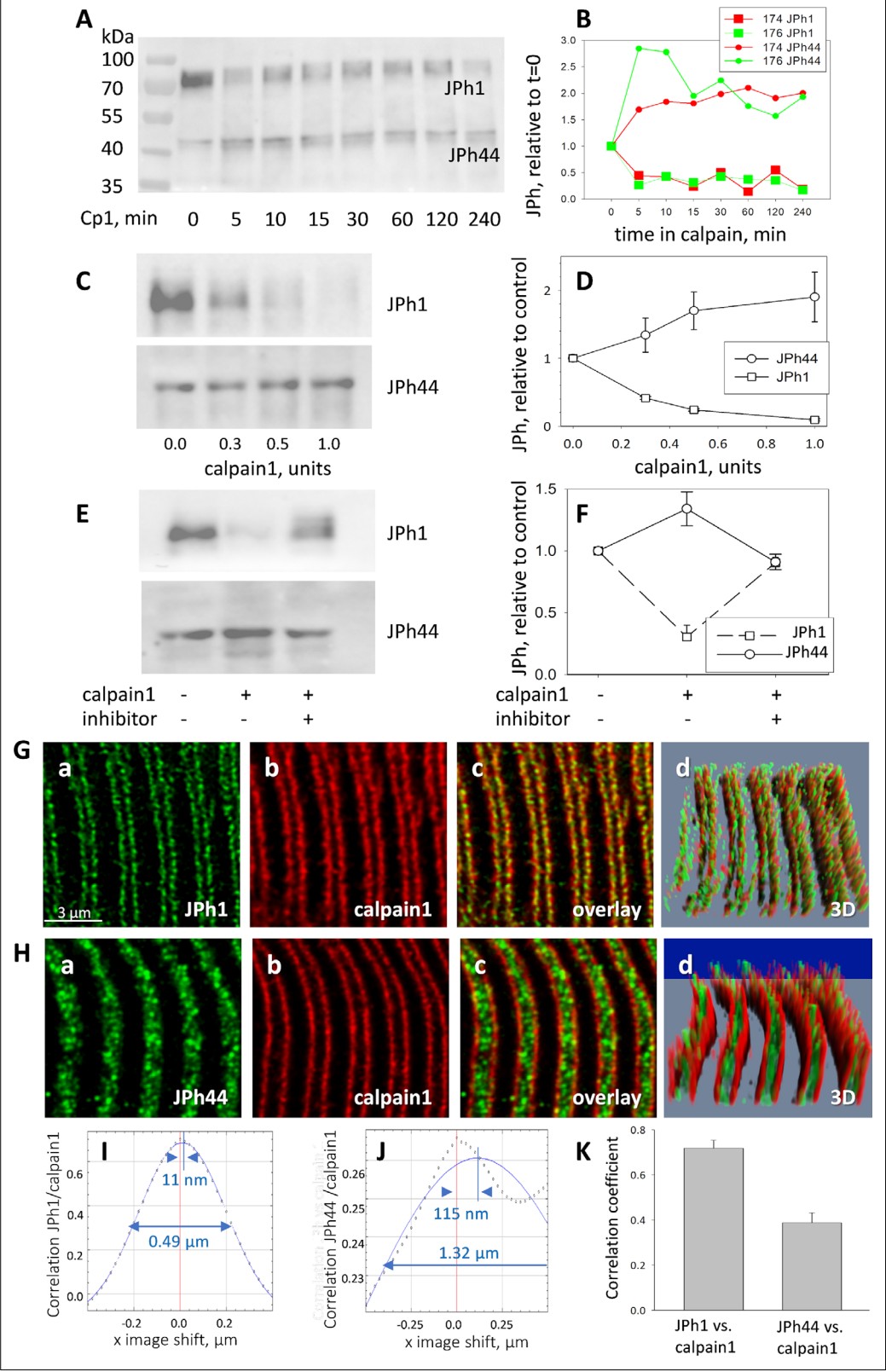

**Figure 5.** Calpain 1 effects and cellular location. (**A, B**) time dependent changes in content of JPh1 and JPh44 in total protein extracts from patients' muscle, incubated with 0.5 units of calpain1 for different intervals (ab A). (**B**) Quantities normalized to initial value. Colors represent different experiments, on tissue from different individuals (2 replicates). (**C, D**) Effect of different doses of calpain1 incubated 10 min with the same extract (n=5 replicates in

*Figure 5 continued*

extracts from 5 subjects). Bars depict SEM. (**E, F**) Effect of calpain at 0.5 units for 10 min, with or without MDL28170 (n=5). In C and E, ab B was used in the JPh44 blot. (**G, H**) Confocal images of JPh1 and JPh44 (abs A and B), in muscle co-stained for calpain 1. Gd and Hd are 3D representations of the full z-stack, showing movement of JPh44 away from T-SR junctions (the location of calpain) and into the I band. (**I, J**) Van Steensel's plots show large differences in colocalization parameters VS shift and FWHM, quantifying movement of JPh44 away from calpain. (**K**) Correlation between calpain and JPh1 or JPh44 signals. As shown by statistical analysis of replicates in *Table 5*, the differences in colocalization parameters are highly significant. *Data trace.* **A**, **B**, and **E**, **F** in *Calpain files with correlations ER IDL.JNB* (sections 1 and 4). **G**, **H**, experiment 081020 a Series 5 and 081020b Series2, patient ID #183, MHS. IF images from a MHN subject, to show similar colocalization of JPh1 and calpain, are in *Figure 5— figure supplement 1*.

The online version of this article includes the following source data and figure supplement(s) for figure 5:

**Source data 1.** JPh raw blot incubated with JPh abA shown in *Figure 5A*.

**Source data 2.** originating gel for blot shown in *Figure 5A*.

**Source data 3.** JPh1 raw blot in *Figure 5C*.

**Source data 4.** JPh44 raw blot in *Figure 5C*.

**Source data 5.** Originating gel for blots in *Figure 5C*.

**Source data 6.** JPh1 blot in *Figure 5E*.

**Source data 7.** JPh44 blot in *Figure 5E*.

**Source data 8.** Originating gel for *Figure 5E*.

**Figure supplement 1.** Distribution of calpain1 in myofibers from an MHN subject.

*Figure 6* illustrates an additional, frequent observation that we did not pursue further: the cell that expressed the calpain construct (Cell 1) had much greater density of RyR1, consistent with other evidence of a role of the protease in control and promotion of muscle development (e.g. *Goll et al., 1992*).

Together, the results reflect calpain1 activation and autolysis in MHS patients by virtue of their elevated $[Ca^{2+}]_{cyto}$, accompanied by localization of the protease at T-SR junctions, where it cleaves JPh1 to produce JPh44. The activated calpain1 also produces the 40 kDa, activated form of GSK3β.

## Effects of a JPh1 deletion mutant in mouse muscle and a muscle cell line

The presence of the JPh44 fragment in the nucleus suggested that it plays a role in gene transcription. The calpain algorithm locates the likely cleavage site at between R240 and S241. To test the prediction and produce a stand-in for the JPh44 fragment, we generated the plasmid GFP-Δ(1-240) JPh1,

**Table 3.** Colocalization measures of junctophilin vs calpain in patients' myofibers.
p: probabilities in two-tailed t test or * non-parametric difference of medians. *Data trace*: images of experiments 051220, 113020, 081020 a (JPh1) & 081120b & 120121 (JPh44). *Data trace*: Section 3 in *colocalizationJPhandRyR.JNB*.

| | | R | VS shift, nm | | FWHM, μm | ICQ | N, subjects n, stacks |
| | | | longitudinal | transversal | | | |
|---|---|---|---|---|---|---|---|
| JPh1 vs calpain | mean | 0.718 | 31.67 | 44.67 | 0.63 | 0.297 | 3 |
| | median | 0.701 | 28.00 | 39.00 | 0.62 | 0.29 | 9 |
| | sem | 0.036 | 6.41 | 6.18 | 0.02 | 0.012 | |
| JPh44 vs calpain | mean | 0.387 | 66.14 | 103.86 | 1.89 | 0.191 | 2 |
| | median | 0.42 | 60.00 | 89.00 | 1.30 | 0.2 | 7 |
| | sem | 0.045 | 18.19 | 23.24 | 0.77 | 0.028 | |
| | p | | *<0.001 | *0.151 | *0.08 | 0.02 | 0.007 |

**Table 4.** Effect of FLAG-calpain expression on location of JPh1 antibody A (which detects full-size and 44 kDa fragment) in patient-derived myotubes.

Location evaluated as ratio of nuclear/cytosolic density of antibody signal. *Data trace*: Section 5 in *colocalizationJPhandRyR.JNB*.

|  | With FLAG-calpain | Reference |
|---|---|---|
| Average | 3.7411 | 1.1816 |
| Median | 3.4118 | 1.1839 |
| SEM | 0.3292 | 0.0923 |
| p | <0.001 |  |
| Subjects, z-stacks | 3, 19 | 4, 18 |

coding for the N-terminal fusion of GFP with the human JPh1 deletion variant that starts at S241, and studied its expression in FDB muscles of adult mice transfected by electroporation. Panels A and B in *Figure 7* compare expression of GFP-JPh1, the full-size construct, with the GFP-tagged deletion variant in myofibers co-stained for RyR1. While GFP-JPh1 remained near RyRs, in triad junctions, the variant moved away, into the I band, in a manner reminiscent of the movement of JPh44 (compare with *Figure 2A and B*). The movement away from triads (marked by RyR1) was quantified by colocalization metrics (*Table 7*). Panels 7 C and D demonstrate intranuclear localization of the deletion variant, reaching concentrations that saturate the light detector at the excitation intensities required to reveal the sarcomeric expression of the protein (e.g. 7 C).

The similarity of movements, cellular location and antibody reactivity between the native JPh44 and the exogenous deletion mutant are consistent with a tentative identification of JPh44 as the C-terminal piece of JPh1 with N terminus at S241. The sole alternative sequence consistent with the predictive algorithm, starting at S234, would only differ for having 7 additional residues at the N terminus. Based on this likely identification, we used GFP-JPh1 Δ(1-240) as surrogate to uncover the effects of the native JPh44 fragment on gene transcription and translation.

To this end, GFP Δ(1-240) JPh1 was transfected into C2C12 cells, where its expression was compared with that of GFP-JPh1, using antibodies A and B (*Figure 8—figure supplement 1*). The deletion construct expressed well and could be found inside nuclei (*Figure 8A*). Gene expression analysis revealed that, relative to control cells, 121 and 39 genes were respectively repressed or induced (p<0.01) in the transfected myoblasts (*Figure 8B*). KEGG pathway-enrichment analysis identified these genes as intimately related to multiple processes, including regulation of the PI3K-Akt-glucose signaling pathway, energy metabolism, muscle growth and lipid metabolism (8 C). Specifically, transcription of GSK3β and proteins inhibitory of phosphorylation of Akt/protein kinase B (Pck1 (phosphoenolpyruvate carboxykinase 1, PEPCK-C)) (*Gómez-Valadés et al., 2008*), RBP4 (retinol-binding protein 4) (*Graham et al., 2006*), Repin1 (replication initiator 1) (*Kern et al., 2014*), APOC3 (guanidinylated apolipoprotein C3) (*Botteri et al., 2017*), TRAF3 (TNF receptor-associated factor 3) (*Chen et al., 2015*), Ces3a/TGH (carboxylesterase 3 A or triacylglycerol hydrolase) (*Lian et al., 2012*), and NGFR (nerve growth factor receptor) (*Baeza-Raja et al., 2012*; *Baeza-Raja and Akassoglou, 2012*) were inhibited in response to expression of the construct.

These results were validated via qRT-PCR. Consistent with the digital gene expression sequencing analysis, quantification of mRNA content by RT-PCR indicated that the contents of mRNAs for GSK3β, TRAF3, NGFR, and Repin1 were significantly reduced by 50% in cells transfected with GFP-Δ(1-240) JPh1 relative to that in control cells transfected with empty vector (*Figure 8D*). (Transcription of the other antagonists of Akt phosphorylation listed above could not be evaluated due to failure of all primers tested).

As expected from the effects on transcription, we found by Western blot analysis a reduction in GSK3β content in cells transfected with GFP-Δ(1-240) JPh1, by 40% of the content in cells expressing GFP alone (*Figure 8E*). The difference was significant (*Figure 8F*). By high-resolution imaging we found that cells expressing the JPh deletion construct had reduced levels of GSK3β compared to the neighboring non transfected cells, whereas cells expressing the vector coding for GFP and otherwise empty had similar GSK3β levels to its neighboring cells (*Figure 8Hc* and *Table 8*).

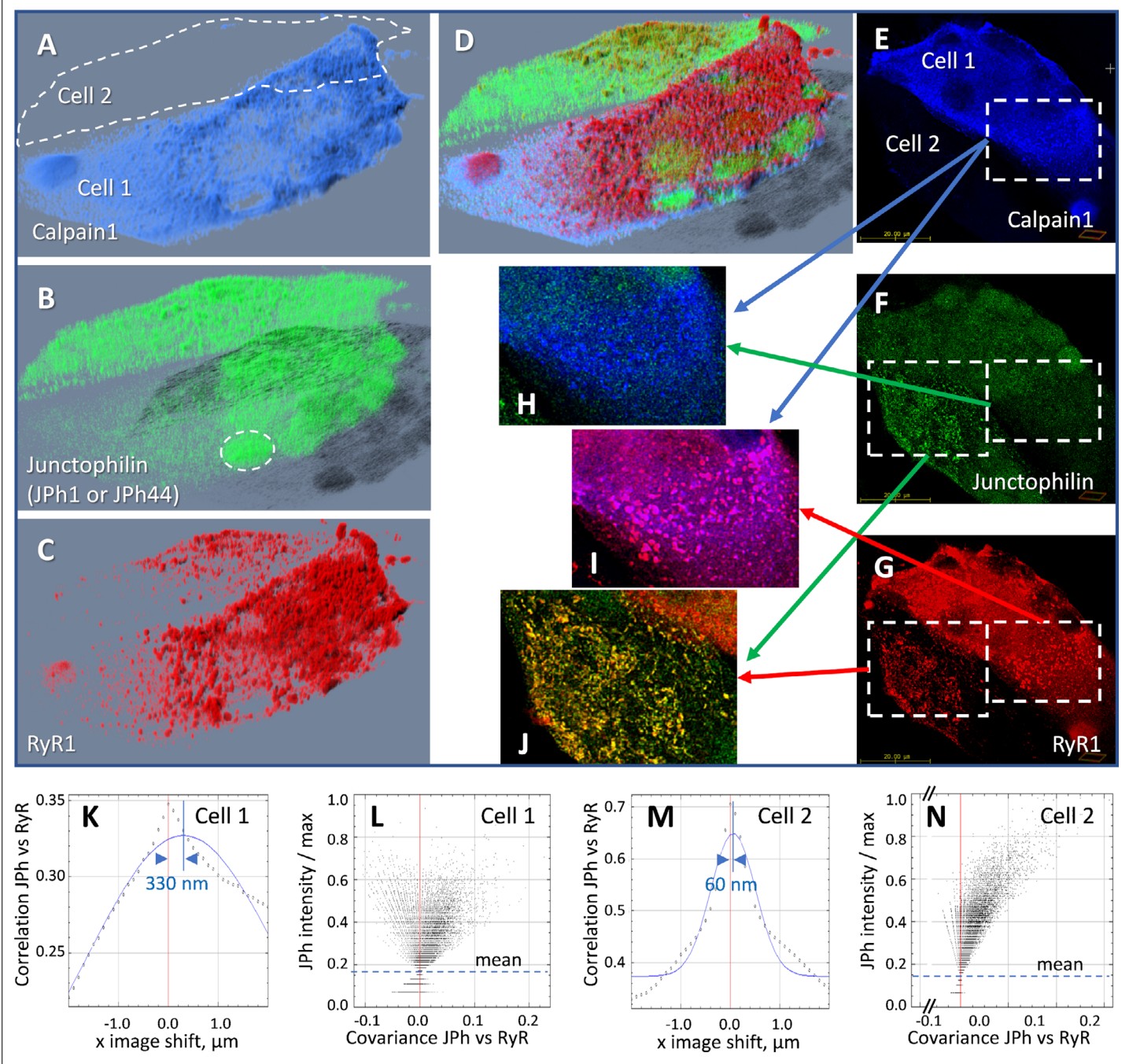

**Figure 6.** Effect of heterologous expression of calpain 1 on human-derived myotubes. (**A-D**) 3D rendering of a z-stack of images from a culture of primary myotubes transfected with FLAG-calpain 1 (panel **A**), co-stained for junctophilin with antibody A, which tags both JPh1 and JPh44 (panel **B**) and RyR1 (panel **C**). While one of the myotubes (Cell 1) expresses calpain 1 abundantly, the other (Cell 2) does not. In Cell 1, junctophilin adopts a fine-grained appearance and occupies nuclei (e.g. ellipse in B), recognizable by the absence of RyR1 (**C, D**). In Cell 2 junctophilin is distributed in the cytoplasm, partly colocalized with RyR1. (**E-G**) An individual x-y image (slice) in the z-stack, with sub-sections magnified and superimposed to illustrate colocalization. (**H**) Calpain colocalizes poorly with ab A in Cell 1, as junctophilin is largely in cleaved, JPh44 form. (**I**) Calpain colocalizes highly with RyR1, forming clusters. (**J**) In Cell 2, which does not express calpain, ab A colocalizes highly with RyR1, indicating full-size JPh1. (**K-N**) Van Steensel's and Li's plots, showing poor colocalization of the junctophilin antibody and RyR1 in Cell 1 (**K, L**), contrasting with that in Cell 2 (**M, N**). Statistics of colocalization measures, comparing cultures with and without calpain, are in *Table 6*. Further evidence of the effect of calpain is provided with *Figure 6—figure supplement 1*. *Data trace*: Experiment 073020La Series 5. Myotubes derived from patient #180, tested as MHN. Data in *ColocalizJp44andRyR.JNB* sections 5 and 11.

The online version of this article includes the following figure supplement(s) for figure 6:

**Figure supplement 1.** Calpain drives nuclear localization of JPh.

**Table 5.** Effect of heterologous calpain on colocalization of JPh (detected with antibody A) and RyR1 in human-derived myotubes.

p calculated by Mann-Whitney Rank Sum Test. *Data trace*: Experiments 073020 a (calpain) & b (reference). 010422 a (calpain). 010422 a (reference). Data in *colocalizationJPhRyR.JNB*, Section #11.

| | N, subjects | n, cells | | R | VS shift, nm | FWHM, µm | ICQ |
|---|---|---|---|---|---|---|---|
| Expressing FLAG-Cp1 | 2 | 13 | avg | 0.44 | 278 | 3.49 | 0.22 |
| | | | median | 0.42 | 130 | 2.24 | 0.23 |
| | | | sem | 0.05 | 127 | 0.96 | 0.03 |
| Reference | 2 | 16 | avg | 0.72 | 44 | 0.58 | 0.30 |
| | | | median | 0.73 | 48 | 0.52 | 0.30 |
| | | | sem | 0.03 | 5 | 0.04 | 0.01 |
| | | | p* | <0.001 | 0.008 | <0.001 | 0.023 |

## Discussion

Starting from the observation of reciprocal differences in content of the structural protein JPh1 and its fragment JPh44 in muscle of patients with chronically elevated cytosolic $Ca^{2+}$, we here identified the pathogenic mechanism leading to these differences. We also demonstrated gene regulatory roles of JPh44, potentially compensatory of the pathogenic consequences of the elevated $[Ca^{2+}]_{cyto}$.

### Junctophilin1, calpain1 and the kinase of glycogen synthase undergo fragmentation in MHS patients

Junctophilin (*Takeshima et al., 2000*) is one of the five proteins deemed essential for functional, skeletal-muscle-style calcium signaling for EC coupling. We now show that a 72 kDa form of JPh1 (the top size detected in our samples) was reduced by ~50% in total protein extracts of muscles of MHS patients (*Figure 1A and C*). A 44 kDa fragment increased in the same patients, albeit in a lesser proportion, a discrepancy that we attribute to proteolysis at other sites in the full-length molecule (*Figure 1B, D and E*).

These changes join a wide-ranging pathogenic alteration of cellular metabolism in these patients, which comprises changes in content, distribution and posttranslational modification of multiple proteins. The observed modification of multiple molecular players of glucose utilization by muscle is the likely cause of the hyperglycemia and diabetes that disproportionally affects MHS patients (*Tammineni et al., 2020*; *Altamirano et al., 2019*). Here we demonstrate a conversion of the original ~47 kDa GSK3β molecule to a~40 kDa form, a truncation that activates the enzyme (*Jin et al., 2015*; *Goñi-Oliver et al., 2007*; *Ma et al., 2012*). Muscle-specific ablation of this protein activates

**Table 6.** Colocalization of JPh and calpain (Cp1) with RyR1 in human-derived myotubes expressing FLAG calpain.

Patient #180 MHN. 073020 a. * p calculated by Mann-Whitney Rank Sum Test. *Data trace*: in *ColocalizJP44 and RyR.JNB* section 9. 073020b. 010422 a.

| Colocalization with RyR1 | | N, subjects | n, cells | R | VS shift, nm | FWHM, µm | ICQ |
|---|---|---|---|---|---|---|---|
| JPh | avg | 2 | 10 | **0.72** | **44** | **0.58** | **0.30** |
| | median | | | 0.73 | 48 | 0.52 | 0.30 |
| | sem | | | 0.03 | 5 | 0.04 | 0.03 |
| Cp1 | avg | 2 | 14 | **0.76** | **126** | **1.13** | **0.34** |
| | median | | | 0.76 | 120 | 0.95 | 0.36 |
| | sem | | | 0.03 | 25 | 0.15 | 0.03 |
| | p* | | | 0.336 | 0.01 | <0.001 | 0.206 |

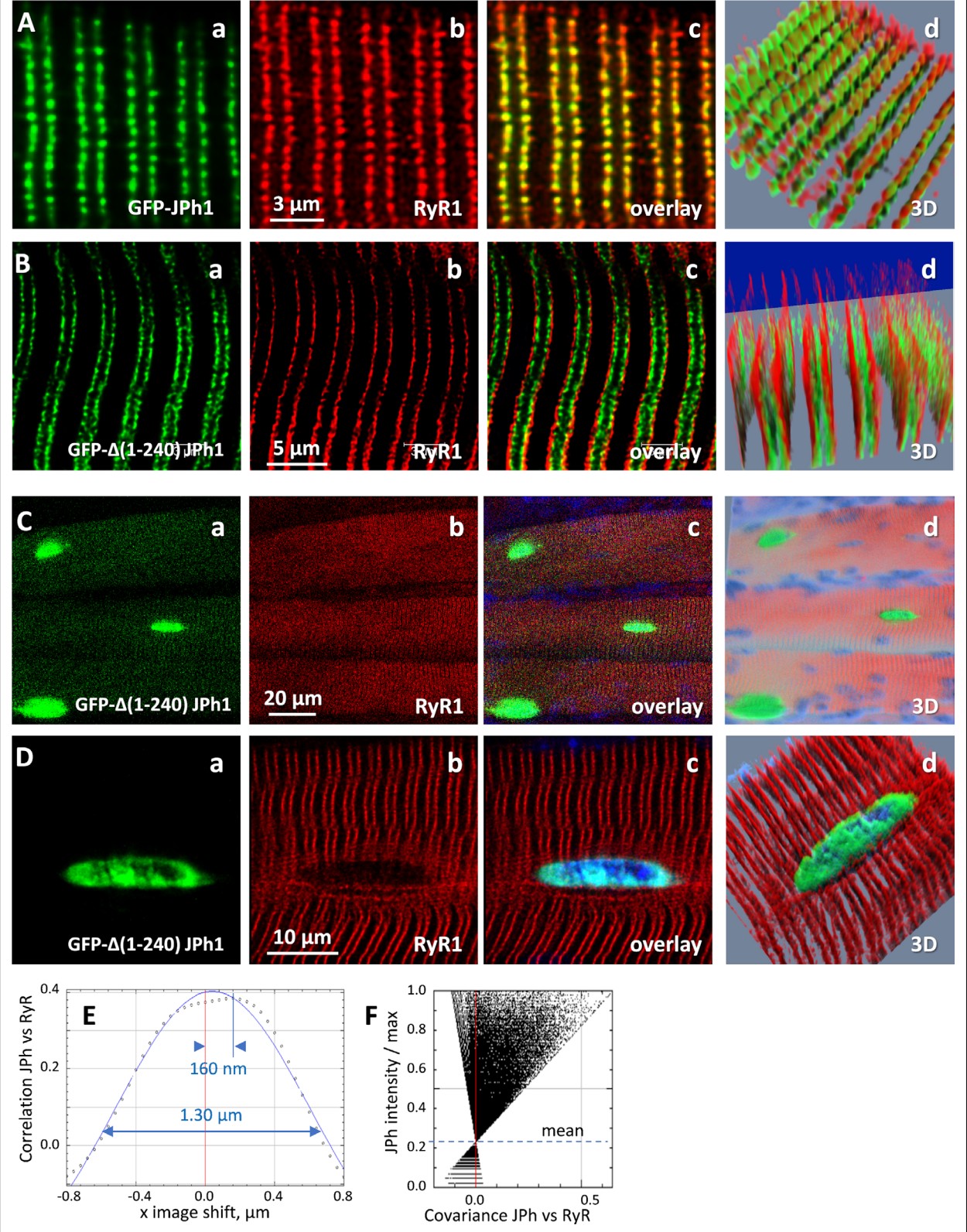

**Figure 7.** Expression of GFP-Δ(1-240) JPH1 and GFP-JPh1 in adult mouse muscle. (**A**) Confocal images of mouse FDB myofibers electroporated with plasmids encoding GFP-tagged, full-length JPh1, co-stained for RyR1. (**B**) Mouse FDB myofibers electroporated with plasmids encoding GFP-Δ(1-240) JPh1. (**C, D**) Demonstrate the nuclear distribution of the protein upon electroporation of GFP-JPh1 Δ(1-240). (**E, F**) Van Steensel and Li plots, showing absence of colocalization between the exogenous construct and RyR1. Statistics of colocalization measures are in *Table 7*. *Data trace*: Source files: **A**, experiment 111220Lb, Series 1. **B-D**, 110920La, Series 1, 8, 9.

**Table 7.** Colocalization of exogenous GFP-Δ(1-240) JPh1 with RyR1 in muscle of adult mice.
For ease of comparison, the table includes values of JPh1 and JPh44 in human muscle, from *Table 1*.
Bottom row: p of no difference between values for fragments vs. JPh1 (Mann Whitney Rank Sum for
all but ICQ). *Data trace*: in *ColocalizJP44andRyR.JNB*.

| | | N subjects | n fibers | R | VS shift, nm | FWHM, |
|---|---|---|---|---|---|---|
| JPh1 | average | 3 | 8 | 0.88 | 16 | 0.42 |
| | median | | | 0.9 | 12 | 0.40 |
| | S.E.M. | | | 0.03 | 3 | 0.07 |
| JPh44 | average | 3 | 10 | 0.31 | 66 | 0.56 |
| | median | | | 0.26 | 54 | 0.50 |
| | S.E.M. | | | 0.04 | 15 | 0.19 |
| | p | | | <0.001 | <0.001 | 0.06 |
| GFP- D(1-240) JPh1 | average | 1 | 6 | 0.54 | 50 | 0.89 |
| | median | | | 0.53 | 36 | 0.80 |
| | S.E.M. | | | 0.09 | 23 | 0.09 |
| | p | | | 0.001 | 0.057 | <0.001 |

glycogen synthase in mice, improving glucose tolerance, correlated with enhanced insulin-stimulated glycogen synthase activation and glycogen deposition (*Patel et al., 2008*). Therefore, we surmise that the activation of GSK3β found in these patients constitutes an additional link in the pathogenic chain that leads from increased $[Ca^{2+}]_{cyto}$ to alteration in muscle utilization of glucose, then to hyperglycemia and diabetes (*Tammineni et al., 2020*; *Altamirano et al., 2019*; *DeFronzo and Tripathy, 2009*).

The present study used techniques for imaging protein localization that, as documented in our earlier work (*Tammineni et al., 2020*), reached a spatial resolution beyond the theoretical optical limit, unprecedented in studies of live muscle. From this vantage point, we demonstrated consequences of cleavage of junctophilin1. While the full-size molecule is located at T-SR junctions (as demonstrated by quantitative measures of colocalization with RyR1; *Figure 2A and C*, and *Table 1*), the fragment JPh44 leaves the junctions and migrates into the I band (*Figure 2B and C*, and *Table 1*).

The fate of JPh44 was further followed on primary myotubes derived from patients' muscle, where it was seen to migrate inside nuclei (*Figure 2E and F*). By contrast, JPh1 remained outside, largely in clusters corresponding to developing T-SR junctions, as revealed by high colocalization with RyR1 (*Figure 2Dd*, Fa, b and *Table 1*). To define the primary sequence of JPh44, the construct (N)GFP-JPh1-FLAG(C) was expressed in myotubes and adult mouse muscle. The consistent presence of the FLAG tag inside nuclei (*Figure 3A–C* and *Table 2*) plus the persistence of the N-terminal GFP in the cytosol, forming clusters analogous to those of JPh1 and RyR1 (*Figure 2*) in myotubes, and staying aligned with junctional triads in myofibers (*Figure 3C*), unambiguously indicate that the doubly-tagged exogenous protein undergoes cleavage similar to the endogenous JPh1, and that the cleaved fraction that enters nuclei -- corresponding to JPh44 -- includes the C terminus of JPh1. The persistence of GFP outside nuclei indicates that the N-terminal fragment cleaved from JPh1 conserves at least some of the features, presumably the MORN motifs, by which JPh1 attaches to T tubule and plasma membranes (*Rossi et al., 2019*). However, the stable location of the N-terminal cleavage fragment of JPh1 at triad junctions is inconsistent with the proposal that assigns the specific triadic location of JPh1 to its ability to dimerize (*Rossi et al., 2019*), as this ability is likely lost in the N-terminal fragment.

## $[Ca^{2+}]_{cyto}$ determines JPh1 fragmentation and relocation

Direct perfusion of permeabilized human myofiber bundles with elevated $Ca^{2+}$ concentrations resulted in reciprocal changes in JPh1 and JPh44 content (*Figure 3D–F*). While the application of $Ca^{2+}$ was only transient, the major, statistically significant changes that resulted are consistent with a causative involvement of the chronically elevated $[Ca^{2+}]_{cyto}$ in the pathogenic process that takes place in MHS muscle. This process included a large increase in JPh44 content in blots of the nuclear fraction

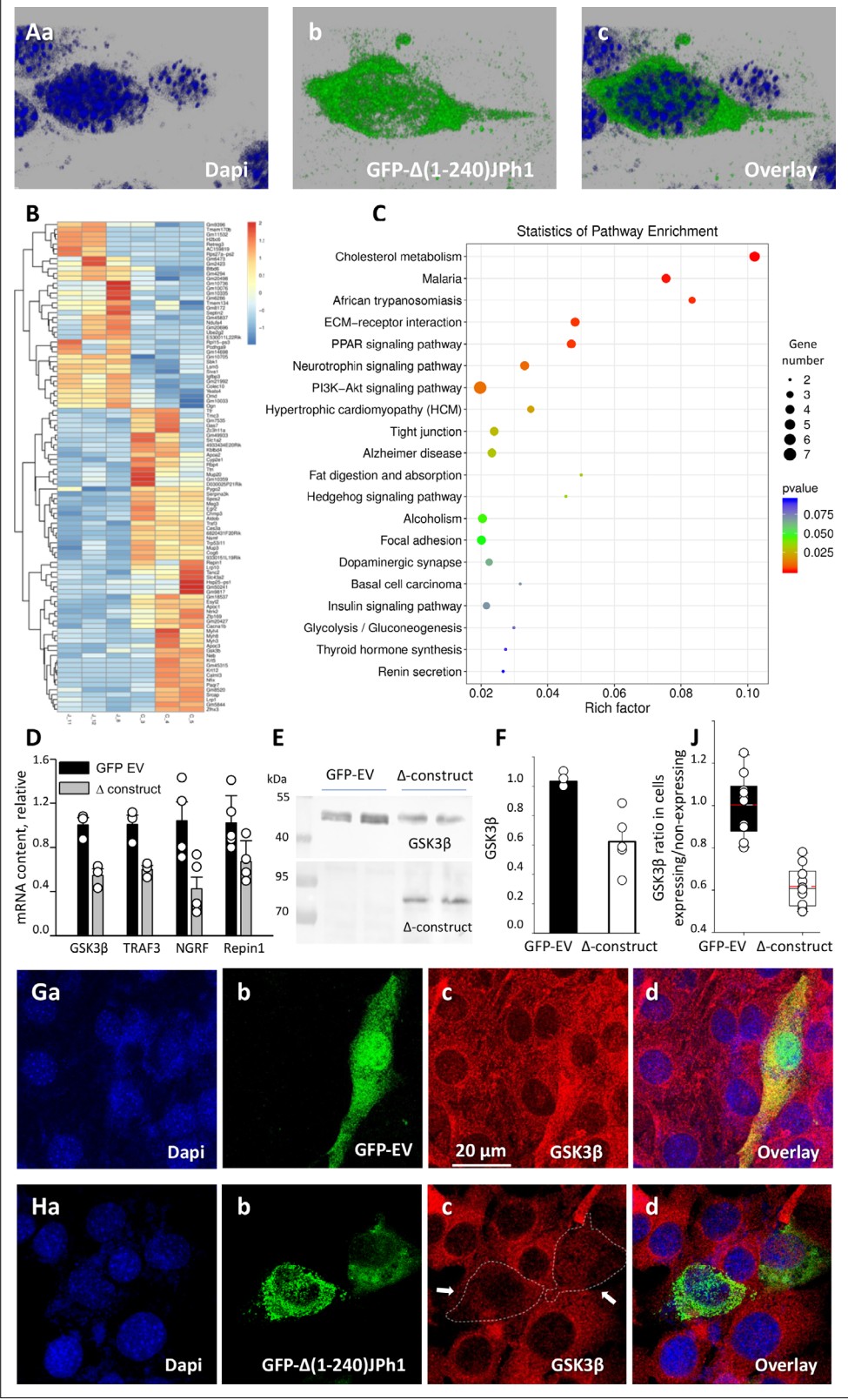

**Figure 8.** Regulation of transcription in C2C12 myoblasts. (**A**) Confocal images of cells transfected with GFP-Δ(1-240) JPh1 vector ('Δ-construct') showing the expressed protein inside nucleus. (**B**) Heat map of significantly altered genes in myoblasts expressing GFP-empty vector or Δ-construct. (**C**) KEGG pathway enrichment analysis of transcripts significantly altered by the construct (analysis by LC Sciences, Houston, TX, USA). (**D**) Validation of

*Figure 8 continued on next page*

*Figure 8 continued*

4 results in B by PCR in our laboratory. Samples from cultures transfected with Δ-construct. Significant differences for GSK3β, p=0.007; TRAF3, p=0.012; NGRF, p*P*=0.015. For Repin1, p=0.053. (**E**) Western blot of GSK3β in total cell extracts from cells expressing GFP-empty vector or Δ construct. (**F**) GSK3β protein signal in WB of panel E. (**G**) Confocal image of GSk3β immuno-fluorescence (red), in C2C12 cells transfected with GFP-empty vector, to compare cells that expressed the marker (green) with those that did not. (**H**) Image for similar comparison of GSK3β content in cultures transfected with Δ construct, showing deficit of GSK3β in expressing cells (green). (**J**) Comparison of ratios of GSK3 β signal in cells showing the GFP tag over those not expressing it. Statistics in *Table 8* shows high significance of the difference. *Data trace*. Source files: **A**, 110121La series 003. **B**, **C**, gene expression profiling data in Dropbox / JPh / manuscript / F9 panels. **D**, in D:/Jph/ manuscript /f9 panels/ *qPCR.JNB*. **E**, uncropped blots and gels in raw western blots and gels.doc. **F**, in *GSK3B levels.JNB* in D:/JPh /manuscript/ F9 panels. **G**, 102921La series9. **H**, 110321 Lb series7. **J**, D:/Jph/manuscript/Fig1/*GSK3B graphs and statistics.JNB*, section 4.

The online version of this article includes the following source data and figure supplement(s) for figure 8:

**Source data 1.** GSK3b raw blot for *Figure 8E*.

**Source data 2.** GFP-Δ(1-240) JPh1 blot for *Figure 8E*.

**Source data 3.** Originating gel for blots in *Figure 8*.

**Figure supplement 1.** Expression of tagged JPh1 and a JPh44 stand-in.

---

extracted from MHS patients muscle (*Figure 3G and H*), which positively correlated with the $[Ca^{2+}]_{cyto}$ measured in myotubes derived from the same patients (*Figure 3I*). This correlation linked a feature of adult muscle with a measure in the derived culture, and notably also applied to JPh localization in patients' myotubes relative to their $[Ca^{2+}]_{cyto}$ values (*Figure 3J* and *Figure 3—videos 1–3*).

## Probable identification of the cleavage site in junctophilin1

The identification of elevated $[Ca^{2+}]_{cyto}$ as mediator of the MHS cellular phenotype pointed at the $Ca^{2+}$-activated calpains as enzymatic agents of JPh proteolysis. An algorithm predictive of calpain cleavage sites within protein sequences (*Liu et al., 2011*) contributed evidence of involvement of calpain1. Among the predicted cleavage sites in JPh1 (*Figure 4B*), the one with the highest score, at R240-S241, is highly conserved in mammals (*Figure 4C*), producing in every ortholog a C-terminal fragment of approximately 46 kDa. This exercise proposes JPh44 as the C-terminal fragment of JPh1 with N

---

**Table 8.** Effect of expression of GFP-Δ(1-240) JPh1 (Δ construct) on density of GSK3β in C2C12 myoblasts.

The reference, in Cols 4–6, is a culture transfected with the empty vector (GFP-EV). The numbers are averages of densities and their ratios calculated individually for 10 and 8 images, respectively for Δ construct and empty vector. The number of expressing cells per image varied between 1 and 3; that of non-expressing cells between 6 and 10. p is the probability of a ratio of 1, that is, no effect of the expression, based on a two-tailed t-test on the sample of 10 or 8 ratios. An alternative t-test (of paired differences in density between non-expressing and expressing areas) yielded a p of no difference = 0.002 and 0.930 respectively for the Δ construct and the empty vector. *Data trace*: in *GSK3B graphs and statistics.JNB*.

| GSK3β density, a. u. | 1 Cells expressing Δ construct | 2 Cells not expressing Δ construct | 3 Ratio col1/ col2 | 4 Cells expressing GFP-EV | 5 Cells not expressing GFP-EV | 6 Ratio col4/col5 |
|---|---|---|---|---|---|---|
| avg | 441 | 729 | 0.62 | 1548 | 1553 | 1.00 |
| median | 353 | 547 | 0.61 | 1513 | 1355 | 1.03 |
| sem | 75.1 | 138 | 0.03 | 268 | 281 | 0.05 |
| N, images | 10 | 10 | 10 | 8 | 8 | 8 |
| n, cells | 15 | 73 | | 11 | 42 | |
| p | | | <0.001 | | | 0.958 |

terminus at S241. The cleavage site S233-S234, also of high probability score, suggests an alternative sequence, identical except for 7 additional aminoacid residues at the N terminus. The extra residues would neither cause a difference in molecular weight detectable by electrophoresis nor be likely to change other physicochemical properties, including putative gene regulatory functions suggested by the nuclear localization. Both cleavage sites predict an N-terminal fragment that contains 6 MORN motifs, a prediction consistent with the observation that the N-terminal fragment of JPh1 stays at T-SR junctions (*Figure 3*). Recent cryo-EM evidence of the tertiary structure of JPh isoforms 1 and 2 (*Yang et al., 2022*) contradicts the conventional view of the helical backbone as spanning the inter-membrane gap (*Lehnart and Wehrens, 2022*), suggesting instead a detailed, presumably tight association of this region with the β sheet formed by the MORNs. If this were the case, a single cut at either site might not be sufficient to allow separation of the distal fragment, suggesting instead that both cuts might be necessary to free JPh44.

A key aspect of this process is the link between elevated $[Ca^{2+}]_{cyto}$ and activation of calpain to directly cleave JPh1. The link was first described in skeletal muscle, starting from the notion that acutely elevated $[Ca^{2+}]_{cyto}$ impairs EC coupling (*Blazev and Lamb, 1999*; *Lamb et al., 1995*), followed by the demonstration of calpain activation and junctophilin proteolysis at $Ca^{2+}$ concentrations in a range ≥500 nM (*Murphy et al., 2013*; *Verburg et al., 2009*). Activation of calpain1 requires only moderate elevation of $[Ca^{2+}]$ (50–300 nM) and includes autolysis of its main subunit, which removes an N-terminal piece to bring the molecular weight from 80 to 76 kDa (*Goll et al., 2003*; *Suzuki et al., 1981*; *Moldoveanu et al., 2002*). We found that in MHS muscle, the fraction of calpain1 in 76 kDa form is about 80% higher than in controls (*Figure 4D and F*). That this piece represents activated protease was affirmed by the high negative correlation between the drop in JPh1 content and the fraction of truncated protease in MHS muscle (*Figure 4H*). A negative correlation of similar significance between calpain truncation and content of the full-length GSK3β is evidence of an additional role of calpain1, namely proteolysis and activation of the GS kinase (*Figure 4I*).

In agreement with the calpain1 predictive algorithm, the protease cleaved the JPh1 protein extracted from human muscle in vitro, in a time- and concentration-dependent manner, to a form with the same apparent molecular weight of the native fragment we call JPh44, an action suppressed by a calpain inhibitor (*Figure 5*).

Additional evidence linking calpain activation to JPh1 cleavage was obtained by its expression in human myotubes. As illustrated in *Figure 6*, heterologous expressed calpain caused a visible increase of JPh44 — distinct from the full-size protein for adopting a fine-grained appearance— which moved massively into nuclei (Cell 1 of panels 6B and D, *Table 4*; see also *Figure 6—figure supplement 1*). Concomitantly, the colocalization of JPh and RyR1 was diminished (*Table 5*). Also consistent with cleavage of JPh1 by the heterologous calpain1 is the high colocalization of the two molecules, contrasting with the low colocalization of JPh44 and the protease (*Table 3*).

Previous studies reported that calpains cleave couplon proteins, including RyR1 (*Place et al., 2015*) and STAC3 (*Ashida et al., 2021*). The present findings show that the protease is at the right location for those actions. Both endogenous calpain1, imaged by immunofluorescence in human muscle (*Figure 5G* and *Figure 5—figure supplement 1*), and FLAG-tagged calpain1 (*Figure 6I*) were found precisely located at triad junctions; the location was confirmed by its high colocalization scores with JPh1 and with RyR1 (*Tables 3 and 6*). The observations support the proposal that the protease is freely diffusible in apo form but binds to cellular structures immediately upon exposure to high $[Ca^{2+}]$ (*van Steensel et al., 1996*). Indeed, the quantitative measures of colocalization with RyR1, especially the spatial shift and Gaussian spread obtained with the Van Steensel method, show that the junctional protein JPh1 has a tighter overlap with RyR1 than calpain1 (*Table 6*), which suggests the coexistence of bound and diffusible forms of the protease.

## From Ca²⁺ dysregulation to hyperglycemia and diabetes

The results reviewed above establish with a high degree of confidence that the reduction in JPh1 content observed in MHS patients is caused by calpain1, activated by excess cytosolic calcium, and that the activated calpain also cleaves the specific kinase of glycogen synthase GSK3β, with inhibitory consequences on glycogen synthesis. This effect, together with the putative cleavage of the regulated glucose transporter GLUT4 (*Otani et al., 2004*), impair glucose utilization by muscle.

In 2019, *Altamirano et al., 2019* first called attention to the high incidence of hyperglycemia and diabetes developing in patients years after they were diagnosed with MHS. Aware that the main proximate cause of insulin resistance and hyperglycemia is failure of glucose processing by muscle (e.g. *DeFronzo and Tripathy, 2009*), we set out to understand the pathogenic pathway linking MHS and diabetes. The question is relevant beyond the MHS syndrome, as a similar dysregulation of $Ca^{2+}$ homeostasis is found in other conditions, including related inheritable diseases with mutations in couplon proteins (*Ríos et al., 2015*; *Dowling et al., 2014*; *Lawal et al., 2020*), DMD (*Edwards et al., 2010*; *Boittin et al., 2006*; *Bellinger et al., 2009*), Exertional and Non-Exertional Heat Stroke (*Leon and Bouchama, 2015*; *Hopkins et al., 1991*) and Statin-related Myotoxicity (*Turner and Pirmohamed, 2019*). We found a wide-ranging alteration of location and phosphorylation of the enzymes that manage storage of glucose as glycogen in muscle, namely glycogen synthase, glycogen phosphorylase and their controlling kinase PhK (*Brushia and Walsh, 1999*), leading to a shift of the glucose ←→ glycogen balance towards glycogenolysis (*Tammineni et al., 2020*). Additionally, we found a decrease in the deployment (translocation) of GLUT4 (*Tammineni et al., 2020*).

The present study adds two nested mechanisms (schematically represented in *Figure 9* together with those previously revealed by *Tammineni et al., 2020*), which operate in the same direction: first, the activation by $Ca^{2+}$ and autolysis of calpain1 (*Figure 4D–F*). Because calpain1 is known to lyse GLUT4 and increase its turnover (*Otani et al., 2004*), its activation is a likely explanation for the observed decrease in translocation of the transporter (*Tammineni et al., 2020*). The second link to diabetes is the observed activation, again by proteolysis mediated by the activated calpain, of GSK3β (*Figure 1F-I*, *Figure 4H, I*). The activated kinase would add its inhibitory effect on GS to that of phosphorylation by the activated PhK (*Tammineni et al., 2020*).

The link between the high activity of GSK3β and diabetes in human skeletal muscle is well established (*Henriksen and Dokken, 2006*), as is the therapeutic potential of GSK3β inhibitors (*Dokken and Henriksen, 2006*; *Maqbool and Hoda, 2017*). It is manifested in our patients by the correlation observed between content of activated kinase and FBS (*Figure 1—figure supplement 3*). Regulation of GSk3β activity by proteolysis was first reported in the brain, where it was found associated with Tau hyperphosphorylation in Alzheimer's disease (*Jin et al., 2015*); the present study provides the first demonstration of its occurrence in skeletal muscle, where it leads to phosphorylation of glycogen synthase, as indicated by the correlation found between GSK3β cleavage and blood sugar.

Taken together, the present observations and those in *Tammineni et al., 2020* identify a multi-lane pathway, where $Ca^{2+}$ activation, proteolysis and phosphorylation, involving at a minimum calpain 1, GSK3β, GS, GP, PhK, and GLUT4, lead from the primary $Ca^{2+}$ dysregulation to hyperthermia and diabetes (*Figure 9*).

## The Ca²⁺-promoted cleavage of junctophilin1 produces an adaptive transcription regulator

Calpains 1 and 2 activate in heart muscle damaged by ischemia (e.g. *Singh et al., 2004*; *Singh et al., 2012*) or in conditions of heart failure, to proteolyze JPh2 (*Murphy et al., 2013*; *Lahiri et al., 2020*; *Guo et al., 2013*). The lysis of JPh2 is associated with structural remodeling, loss of dyadic junctions and deficit of EC coupling function, all expected from the breakage of a structural brace of the junction (*Wu et al., 2014*). Surprisingly, however, L-S Song's group went on to demonstrate that a large N-terminal calpain1-cleaved fragment of JPh2, named JPH2-NTP, enters nuclei, where it regulates transcription of multiple genes, with consequences that oppose the damaging effects of activated proteases (*Guo et al., 2018*). In contrast, *Lahiri et al., 2020* reported the presence of a ~25 kDa C-terminal product of cleavage by calpain2, which was shown to be causative of cellular hypertrophy, rather than compensatory.

The present observations establish similarities and differences between the roles of JPh44 and those of the fragments described in cardiac muscle. The distribution of exogenous doubly tagged JPh1 and its fragments (*Figure 3A and C* and *Table 2*) establishes that JPh44 is a C-terminal fragment. The calpain predictive algorithm, together with the observed distribution of the GFP-tagged Δ(1-240) JPh1, also establish with high likelihood the site of JPh1 cleavage, at either R240-S241 or S233-S234 (*Figure 4A*). In addition to matching the apparent molecular weight of the endogenous JPh44, the C-terminal fragment resulting from cleavage at either point contains two nuclear location sequences, allowing nuclear import of the fragment (*Figure 4A*). Like JPH2-NTP, the fragment will

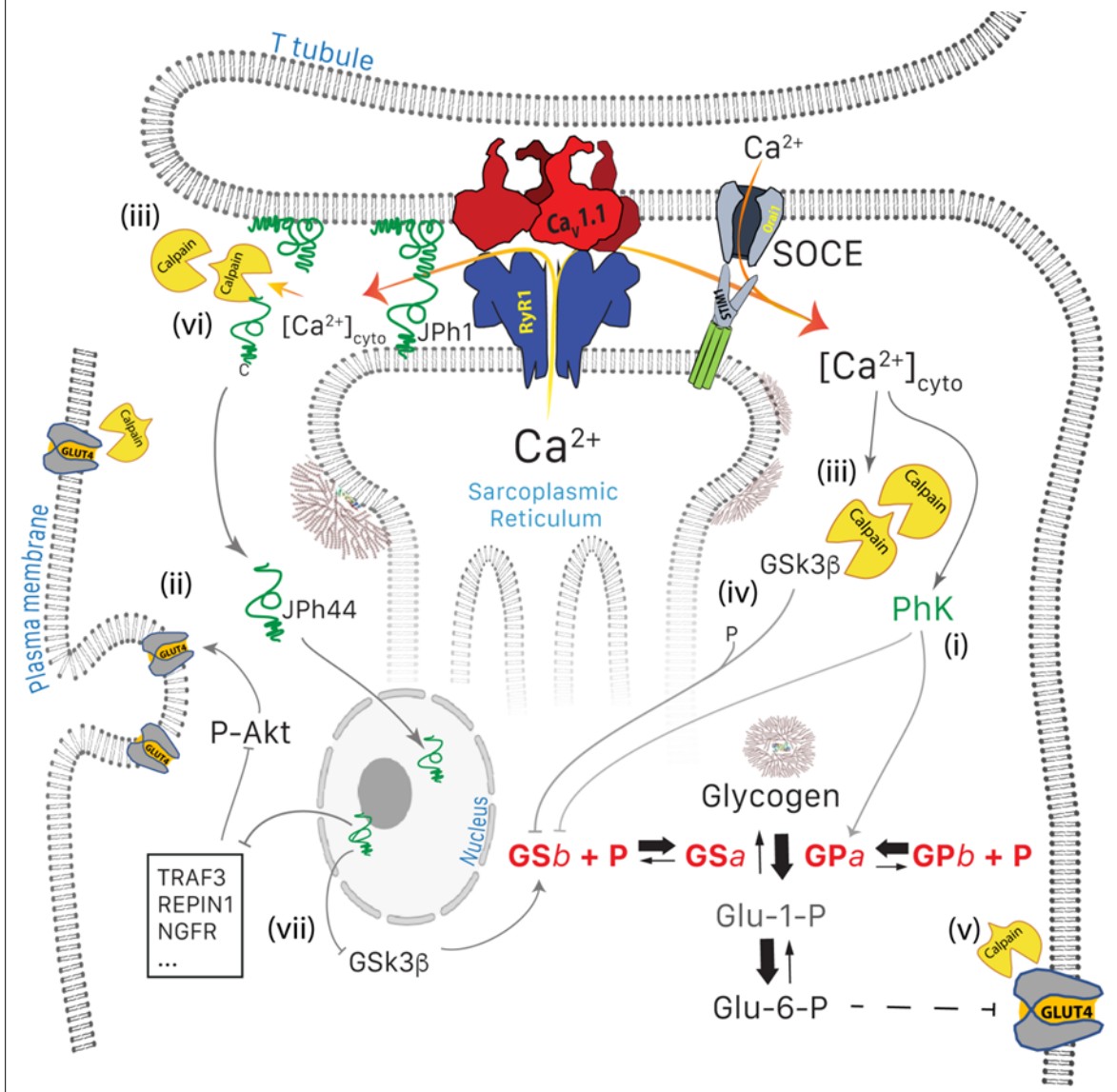

**Figure 9.** Glucose pathways of muscle and their alterations under calcium stress. Glycogen, resident in the intermyofibrillar space in the form of granules or SR-bound, is synthesized from transported glucose or broken down to glucose-1-phosphate (Glc-1-P). Sustained elevation of $[Ca^{2+}]_{cyto}$ induces multiple changes: (**i**) PhK is activated, which increases phosphorylation of GP and GS, to respectively enhance the breakdown and decrease the synthesis of glycogen. (**ii**) Expression and membrane insertion of the glucose transporter GLUT4 decrease (*Place et al., 2015*). (**i**) and (**ii**) contribute to reducing glucose utilization. To these effects, established by *Tammineni et al., 2020*, the present study adds five mechanisms: (**iii**), activation of calpain1, associated with autolysis; (**iv**) cleavage of GSK3β, which activates the kinase to phosphorylate and thereby inhibit GS; (**v**) lysis of GLUT4, which together with (**iv**) depress glucose utilization further, and (**vi**), cleavage of JPh1 to JPh44, which enters nuclei to modulate expression of multiple genes (**vii**). Mechanisms (**i**) to (**v**) contribute to the disease phenotype. While cleavage of JPh (**vi**) should be deleterious to structural stability, the downstream effects of JPh44 are predictive of beneficial consequences, adaptive to the condition of stress.

have an alanine-rich region (ARR), with helix-turn-helix (HTH) structure, characteristic of DNA-binding proteins. These properties establish with reasonable likelihood the primary structure of JPh44 as the C-terminal fragment of JPh1 that starts at S241 or S234. We used this identification to probe directly the possible roles of JPh44 in adult mice muscle and C2C12 myoblasts. When expressed in muscle of adult mice, the GFP-tagged Δ(1-240) JPh1 appeared located similarly as JPh44 in human muscle, largely within the I band and inside nuclei, eschewing triadic junctions and colocalization with its marker, RyR1 (*Figure 7B–D* and *Table 1*).

In C2C12 myoblasts, the construct likewise distributed inside nuclei, as well as in a fine-grained cytosolic form (*Figure 8A* and *Figure 8—figure supplement 1*). Consistent with a gene regulatory

role of JPh44, the transcription profile (obtained by LC Sciences) of myoblasts expressing GFP-tagged Δ(1-240) JPh1 showed significant alterations in multiple pathways (8B, C). Notably, the profile included significantly reduced mRNA levels of GSK3β and at least 9 proteins that act as brakes on the PI3k-AKT pathway, among which TRAF3, Ces3a (*Lian et al., 2012*), Repin1 (*Kern et al., 2014*) and Pck1 (*Gómez-Valadés et al., 2008*) stand out, given their role in muscle metabolism. By quantitative PCR we confirmed the substantial reduction in transcription of four of those genes, as represented schematically in *Figure 9*.

The many signaling pathways regulated by Akt importantly include glucose uptake by muscle and adipocytes. In muscle, Akt isoform 2 is activated via phosphorylation by a chain of events at the plasma membrane, started in response to insulin *Boucher et al., 2014*; this activation promotes translocation of GLUT4 to the membrane, enabling its transport role. Phosphorylation by Akt also inactivates GSK3β, which in turn releases glycogen synthase from inactivation, promoting storage of glucose. Hepatic and skeletal muscle silencing of the four genes named above improved the insulin resistance by inducing phosphorylation of Akt (*Gómez-Valadés et al., 2008*; *Kern et al., 2014*; *Chen et al., 2015*; *Lian et al., 2012*; *Hesselbarth et al., 2017*). Other genes repressed by the exogenous JPh1 fragment JPh44, including Rbp4 (retinol-binding protein 4), Apoc3 (apolipoprotein C-III), MEG3 (maternally expressed gene 3 — a long noncoding RNA) and NGFR contribute to insulin resistance by various actions in other tissues (*Graham et al., 2006*; *Botteri et al., 2017*; *Baeza-Raja et al., 2012*; *Zhu et al., 2019*). Together, the effects on transcription of the exogenous JPh1 fragment are consistent with a beneficial, 'stress-adaptive' role of the endogenous JPh44, similar to that ascribed to JPH2-NTP in stressed myocardium (*Guo et al., 2018*).

In spite of the similarity, there are many differences between the putatively regulatory fragments of the two junctophilin isoforms. Unlike JPH2-NTP, JPh44 loses an N-terminal portion and with it the main stretch of MORN motifs. If the helical backbone were tightly bound to the MORN β sheet, as argued in *Yang et al., 2022*, separation from the N-terminal fragment that includes most MORNs would free the helix for its putative interactions with DNA inside nuclei. Unlike JPH2-NTP, JPh44 retains a second NLS (K588-L614, *Figure 4A*). Therefore, the skeletal muscle fragment appears to have advantages over JPH2-NTP, to leave the junctional location (by losing most T-membrane anchors), and to reach inside nuclei, by having an additional NLS.

## Limitations

The degree of autolysis of calpain in the samples that we analyzed is substantially greater than that reported for muscle tissue processed immediately after excision (*Murphy et al., 2013*; *Murphy et al., 2006a*; *Murphy et al., 2007*). This fact indicates protein degradation in the human samples after excision, and is consistent with the difference in gel migration between our mouse and human samples (*Figure 1—figure supplement 1*). While these indications introduce some uncertainty about the molecular sequence of the JPh fragments of interest, they do not preclude the comparisons between samples from individuals with different diagnosis, as they are all treated in the same manner.

## Ideas and speculations

An additional difference between the MHS condition and the stress of heart failure reported to unleash calpain cleavage of JPh2 is that the cardiac condition is accompanied by structural remodeling with loss of couplons and reduction in JPh2 content (*Wu et al., 2014*; *Zhang et al., 2013*), while no structural remodeling has been reported in muscle of MHS patients. Moreover, studies of $Ca^{2+}$ release function revealed an increase in muscle sensitivity to stimuli (membrane voltage or calcium, e.g., *Figueroa et al., 2019*; *Figueroa et al., 2021*) rather than the major deficits expected to arise from a loss of functional junctions (*Perni et al., 2017*) upon JPh1 loss in the proportion observed here.

A speculative explanation recalls the expression of both JPh1 and 2 in skeletal muscle, both capable of sustaining calcium release function (*Perni and Beam, 2022*). The presence of two molecules that share a function — a redundancy of paralogs — would combine with a different level of redundancy, the presence of multiple copies of the same protein, perhaps polymerized (*Rossi et al., 2019*), to provide a 'sentinel' role. When proteolysis is activated in pathophysiological situations, limited JPh1 cleavage would cause minimal or no deterioration of structure or function, while producing an adaptive gene-regulatory messenger.

## Conclusion

We report that the elevated $[Ca^{2+}]_{cyto}$ in skeletal muscle of individuals with MHS activates calpain1, which associates to T-SR junctions. The protease then cleaves a kinase that inhibits glycogen synthase, an effect that correlates with and probably contributes to an increase in blood glucose concentration. Cleavage of the kinase and that of GLUT4 promote the transition to hyperglycemia and diabetes found in many of these patients. It also cleaves junctophilin1, to produce a C-terminal fragment of ~44 kDa that abandons junctions and enters nuclei. There, the fragment modulates transcription of multiple genes, resulting in effects opposing the deleterious alteration of the glucose utilization pathway. Borrowing a term applied to a fragment of junctophilin2 found to enter nuclei in myocytes of failing hearts (*Guo et al., 2018*), the JPh44 fragment is therefore 'stress-adaptive'.

# Materials and methods

## Key resources table

| Reagent type (species) or resource | Designation | Source or reference | Identifiers | Additional information |
|---|---|---|---|---|
| Antibody | Anti-JPh1'antibodyA' (Rabbit Polyclonal) | Thermo Fisher scientific | Catalog # PA5-52639 | Dilution: IF: (1:200), WB: (1:1000) |
| Antibody | Anti- JPh1'antibody' (Rabbit Polyclonal) | Thermo Fisher scientific | Catalog #40–5100, | Dilution: IF: (1:200), WB: (1:1000) |
| Antibody | Anti- GSK3β (Rabbit Monoclonal) | Cell Signaling Technology | Catalog #9315 | Dilution: WB: 1:1000 |
| Antibody | Anti- calpain1 (Mouse Monoclonal) | Thermo Fisher Scientific | Catalog # MA1-12434 | Dilution: IF: (1:100), WB: (1:1000) |
| Antibody | Anti-Ryr1 (Mouse polyclonal) | Invitrogen | Catalog #MA3-925 | Dilution: IF: 1:200 |
| Antibody | Anti-FLAG (Mouse Monoclonal) | Thermo Fisher Scientific | Catalog #MA1-91878 | Dilution: IF: 1:200 |
| Recombinant DNA reagent | GFP-JPh1-FLAG Plasmid | OriGene Technologies | | |
| Recombinant DNA reagent | GFP-Δ(1-240) JPh1 Plasmid | OriGene | | |
| Recombinant DNA reagent | p3XFlag-CAPN1 plasmid | Addgene | Catalog # #60941 | |
| Biological sample (*Homo-sapiens*) | *Gracilis* muscle | Toronto General Hospital | | |
| Sequence-based reagent | Primers (5' to 3') Fw: ATGGACTGATTATGGACAGGACTG Rev: TCCAGCAGGTCAGCAAAGAAC | IDT | Hprt | |
| Sequence-based reagent | Primers (5' to 3') Fw: GAGCCACTGATTACACGTCCAG Rev: CCAACTGATCCACACCACTGTC | IDT | Gsk3b | |
| Sequence-based reagent | Primers (5' to 3') Fw: CCCATGATCAAACTGCAGAAAC Rev: GCACTCAACTCACTCCTTAGAA | IDT | Traf3 | |

*Continued on next page*

*Continued*

| Reagent type (species) or resource | Designation | Source or reference | Identifiers | Additional information |
|---|---|---|---|---|
| Sequence-based reagent | Primers (5′ to 3′) Fw: GTCTTCAGG CAGAGGAAGAAC Rev: GATTTGCCCT GGTACCTCAA | IDT | Repin1 | |
| Sequence-based reagent | Primers (5′ to 3′) Fw: CTGACAACCT CATTCCTGTCTATT Rev: CTTGCAGCT GTTCCATCTC | IDT | Ngfr | |
| Cell lines | C2C12 Myoblasts | ATCC | CRL-1722 | |
| Protein | Calpain 1 protein from human Erythrocytes | Millipore Sigma | Catalog # 208713 | |
| Drug | Calpain Inhibitor MDL-28170 | Cayman | Item No #14283 | |

## Patients

All patients were recruited and studied in the Malignant Hyperthermia Investigation Unit (MHIU) of the Canadian University Health Network, at Toronto General Hospital, in Toronto, Canada, under protocol 12–5474, last reviewed on 11/08/21. All patients consented to all aspects of the study, including publication. The use of tissue and the language of the informed consent were also approved by the Institutional Review Board of Rush University under protocol 16050502-IRB01. Patients were diagnosed with the CHCT and studied over the last 5 years. Criteria for recruitment of subjects included one or more of the following: a previous adverse anesthetic reaction, family history of MH without a diagnostic MH mutation (https://www.emhg.org), a variant of unknown significance (VUS) in *RYR1* or *CACNA1S*, recurrent exercise- or heat-induced rhabdomyolysis and idiopathic elevation of serum creatine kinase.

The anonymized demographic data of the 44 individuals whose tissues were used for analyses, imaging or development of cultures, is provided in *Supplementary file 1*.

## Primary cultures

Biopsied segments of human *Gracilis* muscle were shipped to Rush University at 4 °C overnight in an optimized solution of composition described in *Figueroa et al., 2019*. Primary cultures, derived from the biopsies as described by *Censier et al., 1998*, were maintained at 37 °C in a humid atmosphere containing 5% $CO_2$. After 8–15 days, cells derived from the explants were transferred to a culture dish for proliferation in a growth medium. Myoblasts were expanded through up to four passages. For calcium concentration measurements in live cells, myoblasts were reseeded on collagen-coated dishes (MatTek, Ashland, MA, USA). For immunostaining experiments, myoblasts were seeded on 13 mm coverslips in 24-well plates. At 70% confluence, the cells were switched to a differentiation medium (DMEM-F12 with 2.5%horse serum). Studies were carried out 5–10 days thereafter, in myotubes showing a similar degree of maturation.w.

## C2C12 cultures

The mouse myogenic C2C12 cell line was obtained from ATCC (http://www.atcc.org/) and cells were used until passage number 20. Myoblasts were cultured in Dulbecco's modified Eagle's medium (DMEM) supplemented with 15% fetal bovine serum and 1% penicillin-streptomycin in a humidified incubator kept at 37 °C and 5% $CO_2$. C2C12 myotubes were obtained by culturing 70% confluent myoblasts in differentiation medium (DMEM, 2% horse serum, 1% penicillin-streptomycin) for at least 4days. Authentication of the cells can rely on that done by the supplier, as most were used within a year of their purchase. We also authenticated C2C12 cell lines based on the morphology observed during experiments. Indeed, these cells turned to multinucleated myotubes upon differentiation, the formation of which is a unique feature of C2C12 cell lines. The cells were tested for Mycoplasma

contamination with a detection kit (MycoStrip, Invivogen) that uses isothermal polymerase chain reaction (PCR) and can detect over 95% of commonly occurring mycoplasma species contaminating cell line. Results were negative.

## Transfections and vectors

Transient transfections of C2C12 myoblasts were performed upon reaching 70% confluence using the K2 Transfection System (Biontex Laboratories GmbH, Munich, Germany) as described by the manufacturer. C2C12 myotubes and biopsy-derived human primary myotubes were transfected with plasmids using lipofectamine 3000 (Thermofisher, Waltham, MA, USA) following the supplier's instructions.

Plasmids used in this study such as empty vector for GFP-JPh1, dual tagged JPh1 plasmid (GFP-JPh1-FLAG), and GFP tagged plasmid with JPh44 translating region (GFP-Δ(1-240) JPh1) were created by OriGene Technologies (Rockville, MD, USA). p3XFlag-CAPN1 was a gift from Yi Zhang (Addgene plasmid # 60941; http://n2t.net/addgene:60941; RRID:Addgene_60941).

## Use of murine muscle

This study was performed in strict accordance with the recommendations in the Guide for the Care and Use of Laboratory Animals of the National Institutes of Health. All the animals were handled according to approved institutional animal care and use committee (IACUC) of Rush University under protocols (#17–035, 18–065 and 21–068.) 6- to 10-week-old mice, *Mus musculus*, of the Black Swiss strain, sourced at Charles River Laboratories (Boston, MA, USA), were used to define the localization of JPh1 and JPh44 in living cells. Hind paws were transfected with plasmid vector for GFP-JPh1, dual tagged JPh1 plasmid (GFP-JPh1-FLAG), and GFP-tagged plasmid with JPh44 translating region (GFP-Δ(1-240) JPh1) as described in *Pouvreau et al., 2007*. Transfection required electroporation via intramuscular electrodes, which was performed under anesthesia. Animals were euthanized and muscles collected and processed for imaging as in *Manno et al., 2017*. Every effort was made to minimize animal suffering and discomfort.

## Diagnosis of malignant hyperthermia susceptibility

Susceptibility to MH was diagnosed at the MHIU following the North American CHCT protocol (*Larach, 1989*). Increases in baseline force in response to caffeine and halothane ($F_C$ and $F_H$) were measured on freshly excised biopsies of *Gracilis* muscle with initial twitch responses that met viability criteria. Three muscle bundles were exposed successively to 0.5-, 1-, 2-, 4-, 8-, and 32 mM caffeine; three separate bundles were exposed to 3% halothane. The threshold response for a positive diagnosis was either $F_H \geq 0.7$ g or $F_C$ (in 2 mM caffeine)$\geq 0.3$ g. Patients were diagnosed as 'MH-negative' (MHN) if the increase in force was below threshold for both agonists, and 'MH-susceptible' (MHS) if at least one exposure exceeded the threshold. While prior work distinguishes the muscles that respond excessively to halothane but not to caffeine (named 'HH') from those that respond to both (the "HS"), in the present work the distinction is not made, and both groups together are classified as MHS.

## Cytosolic calcium concentration

Cytosolic $Ca^{2+}$ concentration, $[Ca^{2+}]$cyto, was monitored in myotubes by shifted excitation and emission ratioing (*Launikonis et al., 2005*) of indo-1 fluorescence as described in *Zhou et al., 2006*. Imaging was by confocal microscopy (scanner TCS SP2; Leica Microsystems; Buffalo Grove, IL, USA), using a 63X, 1.2 numerical-aperture water-immersion objective. $[Ca^{2+}]_{cyto}$ was derived from fluorescence signals (*Figueroa et al., 2012*). Indo-1 was procured from Invitrogen, Waltham, MA, USA.

## Analysis of calpain activity

For in vitro JPh1 cleavage experiments, the whole biopsied muscles from human subjects were homogenized in RIPA lysis buffer (Santa Cruz Biotechnology, Dallas, TX, USA) with protease and phosphatase inhibitors, and the supernatant of centrifugation at 13,000 g for 15 min was used for further process. The cleavage of JPh1 was induced by incubating 100 micrograms of protein supernatants with 0.3–1.0 µg of purified human erythrocyte calpain1 of activity 1 unit/µg (MilliporeSigma, Burlington, MA, USA) at 30 °C for 15 min. Supernatants are treated with or without DMSO dissolved 10 µg calpain inhibitor (MDL28170, Cayman Chemical Co., Ann Arbor, MI, USA). SDS sample buffer was added to stop the reaction.

## Protein fractionation and Western blotting

Human biopsied muscle segments received from the MHIU were quick-frozen for biochemical studies and storage. For measuring total content of proteins in muscle, the tissue was chopped into small pieces in RIPA lysis buffer (Santa Cruz Biotechnology, Dallas, TX, USA) containing protease and phosphatase inhibitors, and homogenized using a Polytron disrupter. The homogenate was centrifuged at 13,000 g for 10 min and supernatant aliquots were stored in liquid nitrogen. Nuclear and cytosolic protein fractions from muscle biopsies were prepared using NE-PER Nuclear and Cytoplasmic Extraction Kit (Catalog. No 78835; ThermoFisher) according to the manufacturer instructions, using a Dounce homogenizer.

Protein content was quantified by the BCA assay (ThermoFisher). Proteins were separated by SDS–polyacrylamide gel electrophoresis, using 10% mini gels and 26-well pre-cast gels (Criterion TGX, Bio-Rad, Hercules, CA, USA), which enable separation in a broad range of molecular weights, and transferred to a nitrocellulose membrane (Bio-Rad). Membranes were blocked at 4.5% blotting grade blocker (Bio-Rad) in PBS and incubated with the primary antibody overnight at 4 °C. Thereafter they were washed in PBS containing 0.1% Tween 20 and incubated in horseradish peroxidase–conjugated anti-mouse or anti-rabbit secondary (Invitrogen, Carlsbad, CA, USA) for 1 hr at room temperature. The blot signals were developed with chemiluminescent substrate (MilliporeSigma) and detected using the Syngene PXi system (Syngene USA Inc, Frederick, MD, USA).

Immunoblot analysis was conducted using the following antibodies (antigen, commercial ID number, concentration, supplier): JPh1 'antibody A', #PA5-52639, 1:1000, ThermoFisher; JPh1 'antibody B', #40–5100, 1:1000, ThermoFisher; GSK3β, #9315, 1:1000, Cell Signaling Technology (Danvers, MA, USA); calpain1, #MA1-12454, 1:1000, ThermoFisher.

Quantitative analysis of Western blots was done with a custom application (written in the IDL platform, Harris Geospatiale, Paris, France; fully described and demonstrated with video in *Tammineni et al., 2020*) that combined information in the blot and the source gel. This method had less variance than the commercial (Syngene) software tools. The content of interest was measured in the blot by the signal mass within a rectangle that enclosed the protein band above a background. A normalization factor, quantifying the sample deposited in the lane, was computed on the gel as the average signal in a large area of the corresponding lane above background. A more advanced version was developed to automatically quantify closely spaced dual bands (as in *Figure 4E*); its use is illustrated in *Figure 4—figure supplement 1*.

Two anti-JPh1 antibodies, A and B, both supplied by ThermoFisher, serendipitously showed different reactivity. When used in Western blots, antibody A (#PA5-52639) reacted approximately equally with the full-size, ~72 kDa molecule and its 44 kDa fragment, while antibody B (#40–5100) marked exclusively JPh44 (*Figure 1—figure supplement 1*).

## Immunostaining of human and mice myofibers, and skeletal muscle cell cultures

Immunofluorescence imaging was done on thin myofiber bundles dissected from human muscle biopsies, FDB muscles from mice, primary human myotubes, C2C12-line myotubes and C2C12 myoblasts. Human or mouse muscles were mounted moderately stretched in relaxing solution, on Sylgard-coated dishes. Relaxing solution was replaced by fixative containing 4% PFA for 20 min. Myotubes or myoblasts on coverslips were washed in 1 X PBS and fixed with 2% PFA for 20 min. Tissues or cell cultures on coverslips were transferred to a 24-well plate and washed three times for 10 min in PBS, then permeabilized with 0.1% Triton X-100 (Sigma) for 30 min at room temperature and blocked in 5% goat serum (Sigma) with slow agitation for 1 hr. The primary antibody was applied overnight at 4 °C with agitation, followed by three PBS washes for 10 min. Fluorescent secondary antibody was applied for 2 hr at room temperature. Dehydrated tissues or cell culture coverslips were mounted with anti-fade medium (Prolong Diamond, ThermoFisher). Immunofluorescence imaging used the following antibodies (antigen, commercial ID number, concentration, supplier): JPh1 'antibody A', #PA5-52639, 1:100, ThermoFisher; JPh1 'antibody B', #40–5100, 1:100, ThermoFisher; calpain1, #MA1-12454, 1:1000, ThermoFisher; Ryr1, #34 C, 1:200, ThermoFisher.

## High-resolution imaging of fluorescence

Immunostained myofibers and cell cultures, as well as live tissues expressing fluorescently tagged proteins were imaged confocally using a Falcon SP8 laser scanning system (Leica Microsystems) with

a 1.2-numerical aperture, water-immersion, 63 x objective. Resolution was enhanced by high sensitivity hybrid GaAsP detectors (HyD, Leica), which allowed low intensity illumination for image averaging with minimum bleaching, optimal confocal pinhole size (below 1 Airy disk), collection of light in extended ranges (e.g. 470–580 nm), and acquisition of z-stacks (vertical sets of x-y images) at oversampled x–y–z intervals. The stacks usually included 40 x-y images at 120 nm z separation and 60 nm x-y pixel size or, for highest resolution imaging, 20 x-y images at 120 nm z and 36 nm x-y pixel size. Dual images were interleaved by line. Most cells were triply stained and correspondingly monitored at (excitation/emission) (405/430–470 nm), (488/500–550 nm) and (555/570–620 nm). The stacks were acquired starting nearest the objective, at or closely outside the lower surface of the myofiber.

Availability of stacks allowed for offline deblurring by a constrained iterative deconvolution algorithm that used all images in the stack (*Voort and Strasters, 1995*; *Agard et al., 1989*) and the point spread function (PSF) of the system, which was determined using 170 nm beads. PSF FWHM was 350 nm in x–y and 480 nm in z. After deblurring, the separation effectively resolved in the x-y plane was approximately 0.1 µm (Supplement 11 in *Tammineni et al., 2020*). The deblurred set was represented or 'rendered' in three dimensions using the 'Simulated Fluorescence Process' (*Messerli et al., 1993*) applied to the full deblurred stack. Determination of the point spread function (PSF) of the imaging system, deblurring, and rendering were done in the HuPro (SVI, Amsterdam, The Netherlands) programming environment.

## Location and colocalization analyses

Location analysis defined densities of protein content within nuclei or in extranuclear areas (named 'cytosol'), by the ratio of integrated fluorescence signal over area in the respective regions. For comparisons among replications the ratio (nuclear/cytosolic) was used, as it was insensitive to inter-preparation variance in expression or staining intensity.

Colocalization (of JPh1, its fragment JPh44, its expressed constructs GFP-JPh1-FLAG, (N)Myc-JPh1 and GFP-Δ(1-240)JPh1, calpain1, its expressed construct (N)FLAG-calpain1 and RyR1) was evaluated by 4 techniques: (*Stern et al., 1997*) the Pearson correlation coefficient, calculated as:

$$R = \frac{\sum_i \left(A_i - \bar{A}_i\right)\left(B_i - \bar{B}_i\right)}{\sqrt{\left(A_i - \bar{A}_i\right)^2 \left(B_i - \bar{B}_i\right)^2}} \tag{1}$$

where $A_i$ and $B_i$ are intensities of two signals in the same pixel i, $\bar{A}_i$ and $\bar{B}_i$ the averages over all pixels, and the summation is extended to all pixels, or a ROI when the entire image is not usable. Subtler, but statistically significant differences in colocalization were detected using the Intensity Correlation Analysis (ICA) introduced by *Li et al., 2004*. This analysis produces a quantitative measure, the Intensity Correlation Quotient, ICQ, a correlation measure with a definition that reduces the influence of heterogeneous staining of protein expression.

$$ICQ = \frac{\sum_i Sign\left[\left(A_i - \bar{A}_i\right)\left(B_i - \bar{B}_i\right)\right]}{N(pixels)} - 0.5 \tag{2}$$

The numerator contributes a 1 for every pixel where both signals are above or below average and a –1 for the situation where the differences are opposite. The ratio is normalized to 1 and the subtraction of 0.5 moves the range to [–0.5, 0.5], with the extremes corresponding to perfect exclusion and perfect colocalization. The ICA approach includes a graph, (with examples in panels *Figure 2Cb*, d), that plots for every pixel the product $\left(A_i - \bar{A}_i\right)\left(B_i - \bar{B}_i\right)$, or 'pixel covariance' of A and B, vs. the intensity of either A or B. The result is strikingly different for cases of colocalization, where the points draw a noisy parabola (e.g. 2Cb) or lack of it (e.g. 2 Cd), where points to the left of the abscissa 0 correspond to mutual exclusion of the two signals.

The conventional correlation analyses have no provision for identifying the anisotropic de-localization of particles, which, if systematic, may reflect an actual association, with systematic displacement. This sensitivity to vectorial displacement is achieved here using an approach of *van Steensel et al., 1996*, which again produces a plot and, in this case, two numerical outputs of interest. The approach (illustrated in *Figure 2* Ca, c) plots the correlation coefficient R between image A and image B shifted in one direction (say, x, defined as the longitudinal direction in myofibers) by variable amounts dx. The 'Van Steensel plot' thus plots R (dx) vs. dx. represented by individual symbols in *Figure 2Ca*. When the two signals are colocalized, the plot is narrow, and centered at dx = 0. If not, the plot displaces its

peak (when the delocalization occurs in a preferred direction) and its width increases. Two numerical quantifiers can be derived on a Gaussian fitted to the points: the abscissa of the maximum (here the 'VS shift') and the FWHM. These quantifiers reveal vectorial aspects of the relationship between the two markers. The VS shift gives a rough measure of distance when the separation of the molecules has some vectorial regularity and sidedness. The FWHM is a measure of dispersion of one marker or both. Combined with the high spatial resolution achieved in our images, it allows to detect effects of interventions that fail to cause significant changes by conventional colocalization measurements.

Colocalization measures and plots were implemented with custom programs written in the IDL platform or with the ImageJ 'plugin' JACoP (*Bolte and Cordelières, 2006*).

## Digital gene expression sequencing

RNA was isolated from GFP-empty-vector-transfected and GFP-Δ(1-240) JPh1-transfected C2C12 myoblasts using the RNeasy plus mini kit (Qiagen, Hilden, Germany), based on manufacturer instructions. Samples were processed by LC Sciences (Houston, TX, USA). Processing included the generation of a library and sequencing of transcripts and their identification in the mouse genome (UCSC mm10). A cDNA library constructed from the pooled RNA from C2C12 cell samples of mouse species was sequenced run with Illumina Novaseq TM 6000 sequence platform. Raw paired-end RNA-seq data were firstly subjected to Cutadapt v1.9 to remove reads with adaptor contamination, low quality bases and undetermined bases. Filtered reads were aligned using HISAT2 to generate alignments tailored for transcript assembler. The mapped reads of each sample were assembled using StringTie with default parameters. Then, all transcriptomes from all samples were merged to reconstruct a comprehensive transcriptome using Gffcompare. After the final transcriptome was generated, StringTie and Ballgown were used to estimate the expression levels of all transcripts and perform expression abundance for mRNAs, by calculating FPKM (fragment per kilobase of transcript per million mapped reads). Differential expression analysis was performed by DESeq2 software between two groups. The genes with the parameter of false discovery rate (FDR) below 0.05 and absolute fold change ≥2 were considered differentially expressed. Differentially expressed genes were then subjected to enrichment analysis of GO functions and KEGG pathways.

## qRT-PCR assays

RNA extraction and real-time quantitative PCR (qPCR): For qPCR, total RNA was isolated from C2C12 myoblasts transfected with GFP-empty vector or GFP-Δ(1-240)JPh1 using the RNeasy-plus mini kit (QIAGEN) according to the manufacturer's instructions. cDNA was synthesized using the iScript gDNA clear cDNA synthesis kit (BIORAD) according to the manufacturer's instructions. The cDNA was then analyzed via real-time qPCR using the SYBR Green qPCR Master (BIORAD) with the ViiA 7 Real-Time PCR System, equipped with a *FAST* 96-well heated block (Applied Biosystems, Life Technologies). Expression levels of HPRT were used as an internal control to normalize the expression of different genes. Quantification was performed by the $2^{-\Delta\Delta Ct}$ method. The qPCR primers used in this study are provided in the 'Resources' table.

## Replications and statistics

With few exceptions, due to limitations in the availability of human samples or their derived cultures, imaging and quantitative analyses were replicated in multiple cells from multiple individuals (biological replicates). In figures and tables, the numbers of individuals (patients, mice) is represented by N and the total numbers of cells by n. When multiple cells derive from multiple individuals, statistical measures are derived by hierarchical (nested) analysis implemented in the R environment (*Sikkel et al., 2017*). Significance of differences of averages or paired differences is established using the two-tailed Student's t test, or, when the distributions of compared measures do not satisfy tests of normality and equal variance, the Mann-Whitney Rank Sum test. Correlation between variables is quantified by the first-order correlation coefficient R; this number is always accompanied by an estimate p of the probability of no correlation, a function of R and the sample size, calculated as described in *Tammineni et al., 2020*. In tables, sample medians are always provided after sample averages, to afford a first idea of the distribution.

The human samples for protein quantification include 12 'normal' and 13 'HS' individuals selected at random from available samples that satisfy size and preservation criteria. The numbers were chosen

for efficiency (as they complete a 25 lane electrophoresis chamber), for consistency with prior work (*Tammineni et al., 2020*), and after analysis that evinced power consistent with study goals (a power of 0.8 for detecting differences of averages of 30%, with standard deviations of 27% at $\alpha$=0.05, is achieved with a sample size of 24, 12 each).

There was no blinding and no exclusion criteria other than adequate size and preservation of the sample — no discoloration or gross damage — evaluated visually upon receipt of the biopsy.

## Availability of raw data

All quantitative raw data, as well as statistical processing are contained in 'JNB' (Sigmaplot) and 'XLSX' (Excel) annotated worksheets, publicly available in Harvard Dataverse as 'Study on junctophilin 1 and calcium stress' at https://dataverse.harvard.edu/dataverse/Junctophilin1_and_calcium included in dataset 'Raw and processed numerical data'. All raw image sets are in the same Dataverse, included in dataset 'Raw image sets for Junctophilin 1 and calcium stress'. For ease of access, a 'data trace' line is added to every figure and table legend, identifying the corresponding raw data in the Dataverse and storage in the Rush University lab. Individual files in the Dataverse contain annotations for traceback to figures and tables in the article.

## Acknowledgements

We are grateful to Prof. Manuel Díaz, for discussion and comments — akin to mentoring — on many aspects of this study, to Prof. Filip Van Petegem for helpful suggestions (notably, that multiple cuts in JPh1 may be needed to release the regulatory fragment), to Prof. Vincenzo Sorrentino, for his generous gift of an N-terminal Myc-tagged junctophilin1 vector, to Profs. Isabelle Marty and Graham Lamb, for detailed comments on the manuscript and its prepublication in bioRxiv. Funding: National Institute of Arthritis and Musculoskeletal and Skin Diseases (R01 AR 071381) to SR, ER, and M Fill (Rush University). National Institute of Arthritis and Musculoskeletal and Skin Diseases (R01 AR 072602) to ER, S L Hamilton, S Jung and F Horrigan (Baylor College of Medicine). National Center for Research Resources (S1055024707) to ER and others. Also supported by matching funds provided by philanthropy contributors to Rush University.

## Additional information

### Funding

| Funder | Grant reference number | Author |
| --- | --- | --- |
| National Institute of Arthritis and Musculoskeletal and Skin Diseases | R01AR071381 | Sheila Riazi<br>Eduardo Rios |
| National Institute of Arthritis and Musculoskeletal and Skin Diseases | R01AR072602 | Eduardo Rios |
| National Institute of General Medical Sciences | R01GM111254 | Eduardo Rios |
| National Center for Research Resources | S1055024707 | Eduardo Rios |

The funders had no role in study design, data collection and interpretation, or the decision to submit the work for publication.

### Author contributions

Eshwar R Tammineni, Conceptualization, Data curation, Software, Validation, Investigation, Methodology, Writing - original draft, Writing - review and editing; Lourdes Figueroa, Data curation, Investigation, Methodology, Writing - review and editing; Carlo Manno, Software, Formal analysis, Visualization, Methodology; Disha Varma, Resources, Validation, Investigation, Methodology; Natalia Kraeva,

Resources, Investigation, Writing - review and editing; Carlos A Ibarra, Conceptualization, Resources, Investigation, Writing - review and editing; Amira Klip, Conceptualization, Supervision, Visualization, Methodology, Writing - review and editing; Sheila Riazi, Conceptualization, Resources, Data curation, Supervision, Funding acquisition, Investigation, Methodology, Writing - review and editing; Eduardo Rios, Conceptualization, Data curation, Software, Formal analysis, Supervision, Funding acquisition, Investigation, Methodology, Writing - original draft, Project administration

Author ORCIDs
Eshwar R Tammineni ⓘ http://orcid.org/0000-0001-7290-9326
Carlos A Ibarra ⓘ http://orcid.org/0000-0001-8898-6772
Amira Klip ⓘ http://orcid.org/0000-0001-7906-0302
Eduardo Rios ⓘ http://orcid.org/0000-0003-0985-8997

Ethics
All patients were recruited and studied in the Malignant Hyperthermia Investigation Unit (MHIU) of the Canadian University Health Network, at Toronto General Hospital, in Toronto, Canada, under protocol 12-5474, last reviewed on 11/08/21. All patients consented to all aspects of the study, including publication. The use of tissue and the language of the informed consent were also approved by the Institutional Review Board of Rush University under protocol 16050502-IRB01.
This study was performed in strict accordance with the recommendations in the Guide for the Care and Use of Laboratory Animals of the National Institutes of Health. All of the animals were handled according to approved institutional animal care and use committee (IACUC) protocols (#17-035, #18-065 and #21-068) of Rush University. The simple surgery (electroporation via intramuscular electrodes) was performed under anesthesia, and every effort was made to minimize suffering and discomfort.

Decision letter and Author response
Decision letter https://doi.org/10.7554/eLife.78874.sa1
Author response https://doi.org/10.7554/eLife.78874.sa2

## Additional files

Supplementary files
• Supplementary file 1. Demographic information for all patients whose samples were analyzed for this study.
• Transparent reporting form
• Source code 1. A program implementing quantitative evaluation of double bands in Western blots.

Data availability
All original data, raw or processed, used in this study, has been made available by publishing it as a Harvard Dataverse containing two large Datasets, with details and location described at end of Materials and Methods. Relevant data on human subjects, including demographics, is provided, fully anonymized, in a supplement. A 'data trace' line, appended to every figure and table legend, allows localization and retrieval of corresponding raw data, from the Dataverse. Individual files in the Dataverse are labeled for traceback to corresponding figures and tables in the manuscript. Uncropped gels and blots are provided as additional files.

The following datasets were generated:

| Author(s) | Year | Dataset title | Dataset URL | Database and Identifier |
|---|---|---|---|---|
| Rios E, Tammineni E, Figueroa L, Manno C, Varma D, Kraeva N, Ibarra C, Riazi S, Klip A | 2022 | Raw and Processed Numerical Data | https://doi.org/10.7910/DVN/YMANVK | Harvard Dataverse, 10.7910/DVN/YMANVK |
| Rios E | 2022 | Raw image sets for 'Junctophilin 1 and calcium stress' | https://doi.org/10.7910/DVN/TDT1HW | Harvard Dataverse, 10.7910/DVN/TDT1HW |

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
