## [Editor Report]

This is an important manuscript that provides evidence in favor of a possible mechanism that explains the lack of glucose utilization in skeletal muscle in pathological conditions (malignant hyperthermia) leading to hyperglycemia. It also describes an interesting compensatory mechanism via junctophilin 1 cleavage and gene expression.

---

## [Decision Letter]

**Decision letter after peer review:**

Thank you for submitting your article "Muscle calcium stress cleaves junctophilin1, unleashing a gene regulatory program predicted to correct glucose dysregulation" for consideration by *eLife*. Your article has been reviewed by 3 peer reviewers, and the evaluation has been overseen by a Reviewing Editor and Richard Aldrich as the Senior Editor. The following individual involved in review of your submission has agreed to reveal their identity: Enrique Jaimovich (Reviewer #2).

Essential revisions:

1) The part of the paper including muscle biopsies from the MHS and MHN individuals appears unfinished. The authors need to include more MHN biopsies to be able to make any valuable comparisons, and for being able to conclude if there are any differences between MHS and MHN (and for respect to the individual donating muscle biopsies). One possibility is to submit the data later.

2) At present in the absence of a solid validation of the ability of antibodies A and B to recognize different band in western blot and in IMF experiments it is difficult to provide a positive answer to the present draft. Also the IMF images need to be carefully revised to provide clear support to the final conclusions.

3). Tammineni and co-authors provide a new interesting role in skeletal muscle for Junctophilin (44 kD segment, JPh44), where it translocates to the nuclei and interferes/affects transcription. Also, the authors shown that Calpain 1 can digest junctophilin to generate the 44 kDa segment. In general, the muscle field know little about how E-C coupling proteins have duals role and interfere/affect gene regulation that subsequently may alter the muscle function and metabolism. This part of the manuscript is solid, informative, and novel. In this paper, the authors also aim to link elevated Jph44 levels in muscle biopsies (myotubes) of patients with malignant hyperthermia susceptible (MHS) (that also exhibit elevated cytosolic calcium with calpain1-associated increases in Jph44 levels) to hyperglycemia in these patients via enhanced GSK3b activity (which inhibits glycogen synthase etc). This part of the manuscript is more speculative and descriptive and lacks proof-of-concept. Also, the authors often compare myotubes of "patients" (unclear whether this always is MHS, or MHN and MHS, needs to be specified) with murine myofibers (and these are presumably healthy controls, but needs to be clarified), and C2C12 myotubes (also control setting). These three vastly different muscle cells from both control and MH individuals makes any type of comparisons problematic and difficult to interpret the results and diverts ones focus from the more direct results regarding Caplain1- Jph44 – translocation – transcriptional changes. There is also data comparisons between myofibers and myotubes from control settings and how they link to MHS/MHN is difficult to interpret. The authors appear to still have access to MHS and MHN (control) samples, and thus to make the conclusions as suggested in the paper the authors need to include both MHS and MHN in all comparisons performed in the study.

Page 2, line 30-31: "On the other hand, LS Song and colleagues (27,30) provided evidence that the intra-nuclear actions of JPh2 31 fragments are "stress-adaptive", partially offsetting the negative consequences of activation of proteolysis". "Stress" can refer to many things, please define stress in this context.

4) Figure 1F, in comparison to other blots, it is not listed in the blot which protein was blotted for.

5) The authors name the JPh antibodies A and B, and then leave it for the reader to find information later in the text about them and how they differ, with the epitope binging to different part of the protein. Would be more reader friendly and informative to earlier in the text give information about the antibodies.

6). In the text the authors describe results as "highly significant" and "marginally significant", please define what p-values it refers to, respectively.

7) Page 4, line 14-16: "The tentative conclusion from this observation is that the cleavage process, which may operate in multiple sites and sequentially on cleaved products, blurs any detailed correspondence between increase in content of a given fragment and disappearance of the full-size protein". Please describe possible fragments that can be generated.

8_ The authors argue that a possible mechanism for that MHS patients develop hyperglycemia and thus at a higher risk of developing type 2 diabetes, is that GSKb3 inhibits glycogen synthase. In this patient cohort (MHS, MHN), what was the fasting glucose levels? Moreover, skeletal muscle is efficient in taking up (e.g., GLUT4) storing (e.g., GS) and utilizing glucose and carbohydrates (glycolysis, TCA) whereas other tissues are lacking this capacity. Hence, how much difference in GSKb3 activity is needed to impact blood glucose levels on a systemic level? Also, what was the activity (phosphorylation) level of GSKb3 in the human samples? On the same note, the authors have previously reported a decrease in GLUT4 content in muscle from MHS individuals. The manuscript would benefit from that the authors acknowledged this finding and thus give a broader understanding of the alterations in glucose homeostasis that is observed in these individuals.

9) his might be a misunderstanding on my side, but I read it as the author hypothesize that elevated [ca^2+^]cyt ◊ Caplain 1 ◊ cleveage of JpH -> larger segments stays in cytosol but the Jph44 translocate to nuclei and impact transcription. Also, MHS subjects have higher Jph44 levels. At the same time, Figure 9 shows that the truncated Jph (mimicking Jph44) results in lower GSK3b expression. What was the protein and phosphorylation levels of this GSK3b? Also, would not a lower GSK3b correspond to less capacity to block GS and thus not be contributing to hyperglycemia as argued above? Please clarify.

10) Page 4, line 22-23 PhK reciprocally activates X… is followed by inhibits Y… Both describes a reduction/inhibition/antagonizing effect of PhK but by using an indirect and direct description, would be more reader friendly to use a direct description for both activities.

11) In the manuscript, the term "patients" are often used, but not specified if MHN or MHS, please clarify throughout the text.

12) Figure 2. Are these data from control mice? Or MHS-derived material?

13) Figure 3I, were these ca^2+^ measurements performed recently or associated to an earlier publication of the co-authors? Also, the 240 on the x-axis is cropped and x-axis is also partly cropped in 3J?

14) Figure 1 A and B show a western blot of proteins isolated from muscles of MHN and MHS individuals decorated with two different antibodies directed against JPH1. According to the manufacturer, antibody A is directed against the JPH1 protein sequence encompassing amino acids 387 to 512 while antibody B is directed against a no better specified C-terminal region of JPH1. Surprisingly, antibody B appears not to detect the full-length protein in lysates from human muscles, but recognizes only the 44 kDa fragment of JPH1. However, to the best of the reviewer's knowledge, antibody B has been reported by other laboratories to recognize the full-length JPH1 protein.

Thus, is not obvious why here this antibody should recognize only the shorter fragment. In addition, in MHS individuals there is no direct correlation between reduction in the content of the full-length JPH1 protein and appearance of the 44 kDa JPH1fragment, since, as also reported by the authors, no significant difference between MHN and MHS can be observed concerning the amount of the 44 kDa JPH1.

15) The authors show in supplemental Figure 1 that antibody B in total lysates from patients 164 and 167 preferentially recognizes the 44 kDa fragment of JPH1, but also a band at about 100 kDa.

Did the authors consider that the 100 kDa band may correspond to full-length JPH1, as also shown by Murphy et al., 2013 ?

Actually, according to Murphy et al., 2013, the 72 kDa protein detected by antibody A in this report represents a proteolytic fragment of full-length JPH1. Showing the entire membrane decorated with antibodies A and B may help to clarify these points.

Based on the data presented, it is very difficult to accept that antibody A and B have specific selectivity for JPH1 and the 44 kDa fragment of JPH1.

A good control would be to express the full length JPH1 and the JPH44 fragment and perform western blot and immunofluorescence staining with antibodies A and B. In the absence of direct evidence of the specificity of these staining, the Western Blots are not very convincing and the immunofluorescence data even more so.

16) In Figure 2B staining of a nucleus is shown only with antibody B against the 44 kDa JPH1 fragment, while no nucleus stained with antibody A is shown in Figure 2A. Images should all be at the same level of magnification and nuclear staining of nuclei with antibody A should be reported.

In Figure 2Db labeling of JPH1 covers both the nucleus and the cytoplasm, does it mean that JPH1 also goes to the nucleus? One would rather think that background immunofluorescence may provide a confounding staining and authors should be more cautious in interpreting these data.

Images in 2D and 2E refer to primary myotubes derived from patients. The authors show that RyR1 signals co-localizes with full-length JPH1, but not with the 44 kDa fragment, recognized by antibody B. How do the authors establish myotube differentiation?

Control (MHN) myotubes and myofibers should be shown as well. Can they provide images of entire myotubes, possibly captured at different Z planes? How can they exclude they are labeling perinuclear SR membranes? Can the authors show a Western blot analysis of proteins extracted from differentiated MHN and MHS myotubes showing immunoreactivity with the two antibodies?

17) Figure 3 A-C. The authors show images of a full-length JPH1 tagged with GFP at the N-terminus and FLAG at the C-terminus. In Figure 3Ad and Cd the Flag signal is all over the cytoplasm and the nuclei: since these are normal mouse cells and fibers, it is surprising that the FLAG signal is in the nuclei with an intensity of signal higher than in patient's muscle.

Can the authors supply images of entire myotubes, possibly captured in different Z planes? How can they distinguish between the cleaved and uncleaved JPH1 signals, especially in mouse myofibers, where calpain is supposed not to be so active as in MHS muscle fibers?

18) If the 44 kDa JPH1 fragment contains a transmembrane domain, it is difficult to understand the dual sarcoplasmic reticulum and nuclear localization. To justify this the authors, in the Discussion session, mention a hypothetical vesicular transport of the 44 kDa JPH1 fragment by vesicles. Traffic of proteins to the nucleus usually occurs through the nuclear pores and does not require vesicles. Even if diffusion from the SR membrane to the nuclear envelope occurs, the protein should remain in the compartment of the membrane envelope. There is no established evidence to support such an unusual movement inside the cells.

19) In Figure 5, the authors show the effect of Calpain1 on the full-length and 44 kDa JPH1 fragment in muscles from MHS patients. Can the authors repeat the same analysis on recombinant JPH1 tagged with GFP and FLAG? Can the authors provide images from MHN muscle fibers stained with JPH1 and Calpain1.

20) In Figure 6, the authors show images of MHS derived myotubes transfected with FLAG Calpain1 and compare the distribution of endogenous JPH1 and RYR1 in two cells, one expressing FLAG Calpain1 (cell1) and one not expressing the recombinant protein. They state that cell1 shows a strong signal of JPH1 in the nucleus, while this is not observed in cell2. Nevertheless, it is not clear where the nucleus is located within cell2 since the distribution of JPH1 is homogeneous across the cell. Can the authors show a different cell?

21) In Figure 7, panels Bb and Db: nuclei appear to stain positive for JPH1. It is not clear why in panels Ac, Bc they show a RYR1 staining while in panels Cc and Dc they show N-myc staining. The differential localization to nuclei appears rather poor also in these panels.

22) The strong nuclear staining in Figure 8, panels C and D is very different from the staining observed in Figure 2 and Figure 3. Transfection should not change the ratio between nuclear and cytoplasmic distribution.

23) Transcriptomic analysis was performed on a population of C2C12 transfected with the GFP-∆(1-240)JPH1 plasmid. Given that the muscles from MHS patients are available, the reviewer suggest performing transcriptomic analysis on muscle fibers from MHS patients as well.

---

## [Author Response]

Essential revisions:1) The part of the paper including muscle biopsies from the MHS and MHN individuals appears unfinished. The authors need to include more MHN biopsies to be able to make any valuable comparisons, and for being able to conclude if there are any differences between MHS and MHN (and for respect to the individual donating muscle biopsies). One possibility is to submit the data later.

We are limited by the availability of patients. The ones included in the study were recruited over several years. The current level of recruitment is approximately 25 per year; We plan to submit additional data as we get them. The recruitment continues but at the present rate --~25 per year-- for the substantial change in sampling errors that the reviewers request, we will need many months. Independently of this purpose, the significance of the findings is generally high. We admit that the tests’ power is not satisfactory, but it should not affect the reliability of the positive (significant) results.

2) At present in the absence of a solid validation of the ability of antibodies A and B to recognize different band in western blot and in IMF experiments it is difficult to provide a positive answer to the present draft. Also the IMF images need to be carefully revised to provide clear support to the final conclusions.

We performed multiple additional tests relevant to the validation and specificity of these abs, which support the way in which we use them. To summarize: in agreement with criticism from Rev. 3, we acknowledge that ab A reacts to both JPh1 and the 44 kDa fragment. In fact, new comparisons of signals and analysis of their colocalization in cells expressing separately the two proteins (suggested by the referee), show that the ab reacts equally to both the full sequence JPh1 and the C-terminal fragment that starts at residue #241 (our “stand-in” for JPh44). By contrast, and confirming our original view, the experiments confirm the specificity of ab B for the fragment. Detailed results and interpretation are provided in this document with responses to items 14 and 15.

3) Tammineni and co-authors provide a new interesting role in skeletal muscle for Junctophilin (44 kD segment, JPh44), where it translocates to the nuclei and interferes/affects transcription. Also, the authors shown that Calpain 1 can digest junctophilin to generate the 44 kDa segment. In general, the muscle field know little about how E-C coupling proteins have duals role and interfere/affect gene regulation that subsequently may alter the muscle function and metabolism. This part of the manuscript is solid, informative, and novel. In this paper, the authors also aim to link elevated Jph44 levels in muscle biopsies (myotubes) of patients with malignant hyperthermia susceptible (MHS) (that also exhibit elevated cytosolic calcium with calpain1-associated increases in Jph44 levels) to hyperglycemia in these patients via enhanced GSK3b activity (which inhibits glycogen synthase etc). This part of the manuscript is more speculative and descriptive and lacks proof-of-concept.

The work describes a response to high Ca that reinforces a brake in glycogen synthase activation. In synergy with the mechanism that we previously reported (*eLife* 2020) it constitutes evidence for a double hit on glycogen synthesis*.* Also, we uncover lysis of a crucial structural protein, with at least one major fragment migrating into nuclei. In addition, the associated changes in expression of multiple genes suggest an evolving attempt for a compensatory response. We posit that the potential compensatory response may not suffice, however, to offset the net reduction in glucose storage by the muscle. On balance, this could result in a net contribution to the observed hyperglycemia in the MHS patients.

This analysis is now included in the manuscript. We hope the reviewer agrees that the hit and attempt at adaptation paradigm is common in pathophysiological responses to abnormal events (in this case, an elevation in cytosolic Ca).

Also, the authors often compare myotubes of "patients" (unclear whether this always is MHS, or MHN and MHS, needs to be specified) with murine myofibers (and these are presumably healthy controls, but needs to be clarified), and C2C12 myotubes (also control setting).

All of the above have been clarified.

These three vastly different muscle cells from both control and MH individuals makes any type of comparisons problematic and difficult to interpret the results and diverts ones focus from the more direct results regarding Caplain1- Jph44 – translocation – transcriptional changes.

In the current study, comparisons of MHN vs MHS muscles were limited to possible molecular changes of JPh1, JPh44, GSK3b, and calpain. Animal muscles and cultures were used to evaluate mechanistic links with manipulations not doable on human muscle biopsies. Having stated this, we agree that the multiplicity of approaches may not add clarity and have removed from the manuscript the studies using extracellular application of calpain (and removed original Figure 7 and Tables 7, 8 and 9), while strengthening other evidence with additional studies illustrated with supplemental figures.

There is also data comparisons between myofibers and myotubes from control settings and how they link to MHS/MHN is difficult to interpret. The authors appear to still have access to MHS and MHN (control) samples, and thus to make the conclusions as suggested in the paper the authors need to include both MHS and MHN in all comparisons performed in the study.Page 2, line 30-31: "On the other hand, LS Song and colleagues (27,30) provided evidence that the intra-nuclear actions of JPh2 31 fragments are "stress-adaptive", partially offsetting the negative consequences of activation of proteolysis". "Stress" can refer to many things, please define stress in this context.

This suggestion may stem from our failure to communicate the different purpose of some experiments. Thus, while we had an overarching goal of establishing pathogenesis and differences in MHS, the exploration of basic processes often did not require differentiating a setting (MHS vs MHN), but simply establishing test vs control conditions. The relevance of the diagnostic condition is especially limited in the case of MHS; the physical signs of which are so modest that it is not even referred to as a disease but as just a “condition”.

In any case, 3D videos of myotubes from one of the MHN and one of the MHS patients included in the comparison illustrated with Figure 3J are included as supplemental material to that figure (page 21 line 9).

4) Figure 1F, in comparison to other blots, it is not listed in the blot which protein was blotted for.

We added a “GSK3b” label to Figure 1F blot.

5) The authors name the JPh antibodies A and B, and then leave it for the reader to find information later in the text about them and how they differ, with the epitope binging to different part of the protein. Would be more reader friendly and informative to earlier in the text give information about the antibodies.

We identify the antibodies in Methods, and now include the following in Results, first two paragraphs:

“…The immunoblot, stained using a polyclonal antibody raised against a human JPh1 immunogen (residues 387-512), referred to here as “A”, is shown in Figure 1A… …A blot from a different gel, stained with a different anti-JPh1 antibody (“B”, raised against a more distal mouse JPh1 immunogen, 509-622), …”

6) In the text the authors describe results as "highly significant" and "marginally significant", please define what p-values it refers to, respectively.

We now include p values everywhere significance is mentioned.

7) Page 4, line 14-16: "The tentative conclusion from this observation is that the cleavage process, which may operate in multiple sites and sequentially on cleaved products, blurs any detailed correspondence between increase in content of a given fragment and disappearance of the full-size protein". Please describe possible fragments that can be generated.

The calpain cleavage algorithm predicts multiple sites with high probability in the full-length JPh1 sequence, shown in the diagram of figure 4. Just those favored sites predict a large number of fragments. The cleavage probabilities in situ will be altered further as lysis progresses, eliciting conformational changes that may lead to site exposure or masking in the newly created fragments. In such scenarios, multiple fragments may form at different rates, both from full-length JPh1 and from secondary cleavage. A reasonably complete determination of fragments would require an in-depth experimental study, exemplified by that of Wang et al. (2021) on JPh2. For the reasons above we have limited our consideration to the fragments that we do find in substantial amounts.

8) The authors argue that a possible mechanism for that MHS patients develop hyperglycemia and thus at a higher risk of developing type 2 diabetes, is that GSKb3 inhibits glycogen synthase. In this patient cohort (MHS, MHN), what was the fasting glucose levels?

The levels are shown in supplemental figure 3 to Figure 1. There is a positive correlation between FBS and both the content of GSK3β cleaved to 40 kDa and the GSK3β 40/47 kDa ratio.

Moreover, skeletal muscle is efficient in taking up (e.g., GLUT4) storing (e.g., GS) and utilizing glucose and carbohydrates (glycolysis, TCA) whereas other tissues are lacking this capacity. Hence, how much difference in GSKb3 activity is needed to impact blood glucose levels on a systemic level?

It is difficult to assign a definite value to the activity of GSK3b that impacts blood glucose levels. In fact, we doubt that such a threshold can be defined. GSK3b activity is controlled by multiple factors including phosphorylation at different sites and proteolytic cleavage of the protein. The control of GS is in turn also multiple and complex. Then the control of glycogen glucose balance is controlled at various points. Add to that the many other ways in which utilization of glucose by muscle is regulated, starting with the uptake transport mechanisms and the concurrent utilization by other tissues. (We believe this multicausal genesis of hyperglycemia explains the delayed onset of it, and diabetes, relative to the Ca stress and MHS condition at the level of muscle.) Therefore, we prefer to stress the actual positive correlation (illustrated in the supplemental figure 3).

Also, what was the activity (phosphorylation) level of GSKb3 in the human samples?

GSK3b has multiple activity-driven phosphorylation sites both on C and N terminal ends. The picture of phosphorylation is complicated by active regulatory cleavage at both N and C terminals. An in-depth study, outside the scope of the present work would be required to answer your question, as it would have to (1) determine the sites of GSK3b cleavage in MHS subjects and (2) how different phosphorylation sites on both ends occur in full-length vs truncated GSk3b.

On the same note, the authors have previously reported a decrease in GLUT4 content in muscle from MHS individuals. The manuscript would benefit from that the authors acknowledged this finding and thus give a broader understanding of the alterations in glucose homeostasis that is observed in these individuals.

The finding is mentioned on page 22, lines 20 and 30, and 23, line 7. It is also included in the diagram of figure 10 (now 9) and its legend.

9) his might be a misunderstanding on my side, but I read it as the author hypothesize that elevated [ca^2+^]cyt ◊ Caplain 1 ◊ cleveage of JpH -> larger segments stays in cytosol but the Jph44 translocate to nuclei and impact transcription. Also, MHS subjects have higher Jph44 levels. At the same time, Figure 9 shows that the truncated Jph (mimicking Jph44) results in lower GSK3b expression. What was the protein and phosphorylation levels of this GSK3b?

Please see our response to item 3 (page 1), where we address carefully this issue. As stated there, the effects of chronically elevated calcium are primarily deleterious (some were described in Tammineni et al. *eLife* 2020; others, described here, include the lytic activation of GSK3b and the cleavage of JPh1, leading to reduction in content of the full-size protein). What we additionally find is a full set of gene modulatory effects, upon expression of the exogenous JPh44-mimic, including reduction of GSK3β at the transcriptional and protein levels (Figure 9, now 8 D-J), which we interpret as compensatory for the primary activation of calpain and kinases. While a decrease in GSK3β protein levels in the MHS condition could be expected, its absence can be explained by many other processes associated with the calcium stress and does not invalidate the gene regulatory effect that our study demonstrates.

We did not measure the phosphorylation levels of GSK3b. As we described while responding to item 7, it requires an in-depth study beyond the scope of the present one.

Also, would not a lower GSK3b correspond to less capacity to block GS and thus not be contributing to hyperglycemia as argued above? Please clarify.

GSK3β inhibits GS by phosphorylation; therefore, a lower activity of the kinase should result in potentiation of the synthase. The effect of muscle-specific GSK3β KO in improving glucose homeostasis and insulin sensitivity was studied by Patel et al.(1).

10) Page 4, line 22-23 PhK reciprocally activates X… is followed by inhibits Y… Both describes a reduction/inhibition/antagonizing effect of PhK but by using an indirect and direct description, would be more reader friendly to use a direct description for both activities.

We reworded the confusing sentence to: “Phosphorylase kinase (PhK) reciprocally activates glycogen phosphorylase and inhibits glycogen synthase. In previous work we demonstrated that this enzyme is activated in the muscle of MHS patients(22). Its effects are consistent with an observed shift of the glycogen balance towards…”. (Page 4, line 27).

11) In the manuscript, the term "patients" are often used, but not specified if MHN or MHS, please clarify throughout the text.

In the revised version we have replaced "patients" with “subjects” or “individuals”, while clarifying their diagnosis status wherever it seems relevant.

12) Figure 2. Are these data from control mice? Or MHS-derived material?

The data is from human muscle fibers and biopsy-derived primary myotubes from both MHN and MHS subjects. We also provided additional immunofluorescence “z-stacks” of MHN and MHS subjects as six supplemental videos to Figure 2, based on suggestions in comment 16.

13) Figure 3I, were these ca^2+^ measurements performed recently or associated to an earlier publication of the co-authors?

Some calcium measurements from 3I and 3J were part of data included in the elaboration of a “Calcium Index”, characterizing at cell-level the MHH syndrome. The collected data were presented in a poster at the Annual Meeting of the Biophysical Society. The same data are included (together with other variables for the same subjects), in a manuscript under review in the British Journal of Anaesthesia. Both the manuscript and the poster describe the response of myotubes to halothane and bear no relation with the topics of the present study (except for their general implications for the MH and other syndromes):

Figueroa L, Kraeva N, Manno C, Tammineni E, Sheila R, Rios E. 2021. Differential sensitivity to halothane among subgroups of Malignant Hyperthermia susceptible individuals: insights into dual pathophysiological mechanisms. 65^th^ Biophysical Society Annual Meeting. *Biophysical Journal* 120 (3), 150a-151a.

Also, the 240 on the x-axis is cropped and x-axis is also partly cropped in 3J?

We have corrected the cropping of x axis in panel 3I. 3J seems correct in our version.

14) Figure 1 A and B show a western blot of proteins isolated from muscles of MHN and MHS individuals decorated with two different antibodies directed against JPH1. According to the manufacturer, antibody A is directed against the JPH1 protein sequence encompassing amino acids 387 to 512 while antibody B is directed against a no better specified C-terminal region of JPH1. Surprisingly, antibody B appears not to detect the full-length protein in lysates from human muscles, but recognizes only the 44 kDa fragment of JPH1. However, to the best of the reviewer's knowledge, antibody B has been reported by other laboratories to recognize the full-length JPH1 protein.Thus, is not obvious why here this antibody should recognize only the shorter fragment.

Antibody B is polyclonal, raised against mouse C terminal immunogen, and readily reacts with mice and the rat's full JPh1. Earlier evidence of ab B detecting full-length JPh1 was demonstrated in only murine models. In fact, both full-length and JPh44 in mice extracts are detected by ab B (Author response image 1). The failure of ab B to react with human JPh1 has already been noted by Murphy et al.(3) and confirmed by G.D. Lamb in personal communications.

Dr. Lamb attributes the failure to the murine origin of the immunogen. This fact cannot also justify the reactivity of ab B with the 44 kDa fragment in human extracts. We have performed new experiments to validate and clarify the specificity of ab B, which are included as Figure 1—figure supplement 2 in this revision, and will be discussed further in the answer to item 15.

**Author response image 1. sa2fig1:** ab B reacts with the large protein in mice but not human extracts (see Figure 1—figure supplement 1 in revised version). Note also that the large, presumably full-size JPh1 from mice migrates at higher molecular weight markings than the human protein.

In addition, in MHS individuals there is no direct correlation between reduction in the content of the full-length JPH1 protein and appearance of the 44 kDa JPH1fragment, since, as also reported by the authors, no significant difference between MHN and MHS can be observed concerning the amount of the 44 kDa JPH1.15) The authors show in supplemental Figure 1 that antibody B in total lysates from patients 164 and 167 preferentially recognizes the 44 kDa fragment of JPH1, but also a band at about 100 kDa.Did the authors consider that the 100 kDa band may correspond to full-length JPH1, as also shown by Murphy et al., 2013 ?Actually, according to Murphy et al., 2013, the 72 kDa protein detected by antibody A in this report represents a proteolytic fragment of full-length JPH1.

Multiple apparent molecular weights of JPh1 as calculated by electrophoretic migration have been reported, in both published literature and antibody database web pages (cf. https://www.proteinatlas. org/ENSG00000104369-JPH1/antibody#western_blot). In our gels from human extracts, the largest protein detected as JPh1 migrates at 72-75 kDa (Figure 1, 3, 5), which is consistent with the molecular weight of derived from the sequence (71.7 kDa). See also new Supplemental Figures 1 and 2 to Figure 1 in this revision, documenting that our antibodies mark only one band of molecular weight near the value predicted for the full JPh1 sequence.

Additionally, the 100 kDa band detected by ab B in the original supplemental figure (now superseded) is not more intense than several other weakly stained bands, which suggests that the staining there is non-specific.

We acknowledge in the revised article the possibility of minor degradation of the top-size protein, given the doublet band at 72 kDa in many lanes of text Figure 1A and the difference in migration size between murine and human protein in Response Figure 1. However, and referring to the same figure, in spite of the detection of a massive band at ~70 kDa in the lane with human extract, there is no detection at higher molecular weights, which argues against major degradation of the human protein. In consideration of these concerns, we added a “Limitations” section in Discussion (pages 24, bottom, and 25, top).

Showing the entire membrane decorated with antibodies A and B may help to clarify these points.

We designed an experiment, documented in Supplement 2 to Figure 1, to specifically address this question. The blotted membrane from an electrophoresed gel of human muscle protein was divided in halves, which were separately reacted with antibodies A and B (please note again: the abs were applied on separate halves of a common blot). The gel contained, in duplicate, increasing amounts of protein extract. Ab A detected 72 and 44 kDa bands, with signals roughly proportional to quantity of extract in each lane. Ab B detected JPh44, also commensurate with added extract, and no other protein above the 44 kDa band. (Both antibodies also detected a 30 kDa band of density that varied erratically with lane protein content, the meaning of which could not be interpreted). The experiment, which required a brief pretreatment with calpain to increase the proportion of JPh44, also confirmed that the content of the fragment is much lower than that of JPh1 in most conditions. In recognizing this fact, we are revising our previous assertion that ab A preferentially tags JPh1. An additional experiment shown later will demonstrate a similar affinity for both antigens. We are grateful to the reviewers for noting our error.

The striking and convenient specificity of ab B is also shown in our immunofluorescence images, as ab B did not stain endogenous full-length JPh1 protein at triad junctions or recombinant expression of GFP-JPh1 in *in-vitro* cultures.

Based on the data presented, it is very difficult to accept that antibody A and B have specific selectivity for JPH1 and the 44 kDa fragment of JPH1.

As stated in previous paragraphs, we now accept that ab A does not have specificity for JPh1 over the fragment; on the other hand, the experiments demonstrate clearly the specificity of ab B for JPh44.

A good control would be to express the full length JPH1 and the JPH44 fragment and perform western blot and immunofluorescence staining with antibodies A and B. In the absence of direct evidence of the specificity of these staining, the Western Blots are not very convincing and the immunofluorescence data even more so.

In addition to the uncropped Western blots with ab A and ab B in Supplement 2 to Figure 1, we include the imaging experiment requested. We transfected C2C12 myoblasts with the fusion constructs GFP- JPh1 and GFP-∆(1-240) JPh1 and stained the cultures with ab A and ab B. Images and analysis of colocalization of IF stains and GFP are documented in Supplement Figure 1 to Figure 8. Images and analysis are consistent with our conclusion that ab A tags the endogenous JPh1 and JPh44 (colocalization is quite precisely the same for both ab A signals –Figure 5, panels F, H). As for ab B, some staining was present on cells transfected with the GFP-JPh1 full-size construct, but it was visibly weaker than on cells expressing GFP-JPh44. The analysis correspondingly showed a reduction by ~50% of the colocalization parameters. That there was ab B signal in JPh1-expressing cells does not disprove the specificity of ab B for JPh44, as it could have other explanations (e.g., cleavage by endogenous calpain of the expressed full-size fusion protein, which would produce a 44 kDa fragment poorly localized with GFP). In all, these results establish ab B (an antibody raised against a murine antigen!), as a serendipitous tool to detect the 44 kDa fragment in human muscle.

16) In Figure 2B staining of a nucleus is shown only with antibody B against the 44 kDa JPH1 fragment, while no nucleus stained with antibody A is shown in Figure 2A. Images should all be at the same level of magnification and nuclear staining of nuclei with antibody A should be reported.

Figure 2A was prepared precisely to show how ab A fails to stain nuclei. This is illustrated in the “window” (showing exclusively ab A signal) within the 3D rendering of panel 2Aa. Rendering 2Aa is presented at a magnification that allows visualization of some nuclei, two of which are partly included in the “window”, to show the absence of ab A signal. The other panels in row A are chosen to show details of localization of the antibody, in triads marked by RyR1.

To allay the reviewers’ concern, we now add Figure 2—figure supplement 1, which shows other regions of the same fiber bundle. As requested, ab A and other stains at low magnification are in row A, demonstrating no visible ab A inside nuclei. Rows B and C compare the same signals in two regions: one with conserved structure, the other with triads disrupted and general misalignment, consistent with fiber damage. While we have not confirmed the impression with statistics, intra-nuclear staining with ab A is clearly visible in damaged regions, in this and other biopsies. Ab A staining per se cannot tell the larger protein from its fragment; our conclusion that the intra-nuclear signal corresponds largely to the fragment is based on staining with ab B (the arguments and additional data were presented with items 14 and 15; additional ones are in our response to the next question).

In Figure 2Db labeling of JPH1 covers both the nucleus and the cytoplasm, does it mean that JPH1 also goes to the nucleus? One would rather think that background immunofluorescence may provide a confounding staining and authors should be more cautious in interpreting these data.

As stated above, new WB and image assays (supplemental Figures 2 and 3 to Figure 3, supplemental figure to Figure 8) show that ab A reacts equally well with JPh1 at 72 kDa and the 44 kDa fragment.

In Figure 2Db, ab A detects the protein both outside and inside the nucleus of the primary myotubes. Cytosolic punctate that is colocalized with Ryr1 represents full-length JPh1. Protein reacting with ab A inside nuclei could be 72 kDa JPh1; however, other evidence (e.g., figures 3, 7, Supp. 2 to Figure 1) shows that the 72 kDa protein is excluded from nuclei, while Figures3G and 8 show the 44 kDa protein and its “stand-in” inside nuclei. This suggests that the nuclear staining by ab A observed in primary myotubes represents the 44 kDa fragment rather than being non-specific.

To additionally argue that the ab A signal in myotubes is largely specific, note differences in nuclear staining by ab A between MHN and MHS myotubes in the videos presented in response to Item 3. Also, in Figure 3 J, a high correlation is demonstrated between the ab A nuclear/cytosolic signal ratio and [ca^2+^]_cyto_ in primary myotubes of different subjects, again in favor of a specific nature of the antibody signal. Finally, a full comparison of expression of constructs in C2C!2, (new Figure 8—figure supplement 1) also demonstrates, via colocalization analysis, the specific nature of ab A signals.

Images in 2D and 2E refer to primary myotubes derived from patients. The authors show that RyR1 signals co-localizes with full-length JPH1, but not with the 44 kDa fragment, recognized by antibody B. How do the authors establish myotube differentiation?Control (MHN) myotubes and myofibers should be shown as well. Can they provide images of entire myotubes, possibly captured at different Z planes? How can they exclude they are labeling perinuclear SR membranes?

3D reconstructions (“renderings”) obtained using z stacks of human myotubes and myofibers of both MHN and MHS stained with abs A and B are added as six Supplementary Videos to Figure 2. These renderings, demonstrate that the nuclear signal observed is obtained from the actual nuclear volume rather than the perinuclear membrane. Thanks for the suggestion.

Can the authors show a Western blot analysis of proteins extracted from differentiated MHN and MHS myotubes showing immunoreactivity with the two antibodies?

This is a worthwhile goal. At this point, however, we limited our myotubes studies to IF experiments because they allow us to avoid myoblasts and fibroblasts, which complicate the interpretation of Western blots. We have found that MHN and MHH myotubes are of different size, and evolve differently. The observations, which are ongoing, are beyond the scope of the present manuscript.

17) Figure 3 A-C. The authors show images of a full-length JPH1 tagged with GFP at the N-terminus and FLAG at the C-terminus. In Figure 3Ad and Cd the Flag signal is all over the cytoplasm and the nuclei: since these are normal mouse cells and fibers, it is surprising that the FLAG signal is in the nuclei with an intensity of signal higher than in patient's muscle.

The purpose of the approach illustrated in Figure 3 is to test whether the JPh fragment translocated to the nucleus is C terminal. The intensity with which the fragment is marked changes between cell types; even within the same cell, it can vary from region to region based on the state of the fiber. As shown in Figure 3—figure supplement 1, most nuclei have low or no FLAG signal. As shown in with Figure 2—figure supplement 1, ab A, which tags equally JPh1 and JPh44, appears within nuclei in areas of altered structure and presumably high Ca, but not visibly in regions of normal structure.

Can the authors supply images of entire myotubes, possibly captured in different Z planes? How can they distinguish between the cleaved and uncleaved JPH1 signals, especially in mouse myofibers, where calpain is supposed not to be so active as in MHS muscle fibers?

A perceptive point. For immunoimaging we use thin bundles, which may include damaged fibers. As shown in the previous response, translocation of the C-terminal fragment to the nucleus is variable. Animated “trips” through the full z-stacks from which images in panels A-C were taken are provided as Supplemental videos to Figure 3. The reviewer’s assertion that immunoimaging with ab A does not permit distinguishing between the two JPh1 forms is correct. That distinction can only be made on the basis of ab B staining.

18) If the 44 kDa JPH1 fragment contains a transmembrane domain, it is difficult to understand the dual sarcoplasmic reticulum and nuclear localization. To justify this the authors, in the Discussion session, mention a hypothetical vesicular transport of the 44 kDa JPH1 fragment by vesicles. Traffic of proteins to the nucleus usually occurs through the nuclear pores and does not require vesicles. Even if diffusion from the SR membrane to the nuclear envelope occurs, the protein should remain in the compartment of the membrane envelope. There is no established evidence to support such an unusual movement inside the cells.

In agreement with the argument, we do not have specific evidence for a vesicular pathway, hence we have removed the speculation from the Discussion.

On the question whether the protein that enters nuclei includes the TMD: the nuclear localization signal (NLS) mapper identified two bipartite NLS sequences downstream of the preferred calpain cleavage site in JPh1, as indicated in Figure 4. Despite the presence of a similar TMD in a JPh2 C-terminal fragment, Lahiri et al. (4) showed that a homologous NLS allowed the nuclear translocation of the fragment in cardiomyocytes. Similarly, we believe that NLS sequences in JPh44 may allow its nuclear relocation in spite of the TMD.

19) In Figure 5, the authors show the effect of Calpain1 on the full-length and 44 kDa JPH1 fragment in muscles from MHS patients. Can the authors repeat the same analysis on recombinant JPH1 tagged with GFP and FLAG?

We agree that confirmatory evidence of the calpain effect on dual-tagged recombinant JPh1 would be desirable. However, we think an in-depth study is required to follow up on the number of JPh1 fragments generated by calpain (or by different calpain isoforms) and their positions, similar to the detailed study of JPh2 fragmentation Wang et al. in 2021 (5).

Can the authors provide images from MHN muscle fibers stained with JPH1 and Calpain1.

We provide the images requested as Supplement 1 to Figure 5. As in the original example, calpain partially colocalized with JPh1 at triads, without any visible difference with its distribution.

20) In Figure 6, the authors show images of MHS derived myotubes transfected with FLAG Calpain1 and compare the distribution of endogenous JPH1 and RYR1 in two cells, one expressing FLAG Calpain1 (cell1) and one not expressing the recombinant protein. They state that cell1 shows a strong signal of JPH1 in the nucleus, while this is not observed in cell2. Nevertheless, it is not clear where the nucleus is located within cell2 since the distribution of JPH1 is homogeneous across the cell. Can the authors show a different cell?

The requested set of images, of cell cultures with and without transfection of the calpain construct, are provided as Supplementary Figure 1 to Figure 6. In defense of the original figure, which we would like to keep, we find major value in the visualization side-by-side, in the same images, of two cells with different content of exogenous calpain, one with full expression and the other with none. This concurrence allows a direct comparison, without the usual confounding factors associated with variable staining and other conditions. However, we agree that visibility of nuclei is not the best in the example. In the supplemental figure, nuclei are shown better in the calpain-expressing cells and are specifically stained in the control culture.

21) In Figure 7, panels Bb and Db: nuclei appear to stain positive for JPH1. It is not clear why in panels Ac, Bc they show a RYR1 staining while in panels Cc and Dc they show N-myc staining. The differential localization to nuclei appears rather poor also in these panels.

This criticism, reinforced by commentaries to our prepublication in BioRxiv, indicates that the experiments illustrated in Figure 7 add clutter rather than clarity. In the revised version, we removed references to these experiments, the figure and 3 related tables (original figures 8-10 are now 7-9).

22) The strong nuclear staining in Figure 8, panels C and D is very different from the staining observed in Figure 2 and Figure 3. Transfection should not change the ratio between nuclear and cytoplasmic distribution.

Transfection is an intrusive procedure, which requires production and trafficking of an exogenous protein. This protein, furthermore, is an artificial construct (in this case, a “stand-in”, which adds to the native protein and therefore is akin to overexpression). For the above reasons, we believe that differences in intensity of nuclear staining may obey to multiple causes and should not be especially concerning.

A simple example of the differences is illustrated in Author response image 2. Upon antibody exposure, all nuclei in human myofibers stained for JPh44, as shown in the top row of images. By contrast, mice transfected with GFP-∆(1-240) JPH1 showed the expressed protein in fewer than half of the visible nuclei.

**Author response image 2. sa2fig2:** Top: human muscle stained with ab B and Hoechst nuclear marker. Bottom: mouse muscle transfected with the deletion JPh construct. The GFP signal is imaged as saturation levels, to capture the distribution in the cytosol. *Data trace*: top; 122619L_Series025, patient #169 (HH); bottom; 110920La_Series008. Images from z-stacks corrected for optical dispersion.

23) Transcriptomic analysis was performed on a population of C2C12 transfected with the GFP-∆(1-240)JPH1 plasmid. Given that the muscles from MHS patients are available, the reviewer suggest performing transcriptomic analysis on muscle fibers from MHS patients as well.

Indeed, we are working on strengthening the transcriptomic analysis in several ways. One is to characterize the effect of the deletion protein on a preparation that is further developed than the one we used. To this end, we prioritized the effect in myotubes and in mouse muscle (where the challenge –as illustrated in the previous figure—has been to obtain transfection of a sufficient number of nuclei AND to quantify this fraction -- required for a quantitative interpretation of the results). The transcriptional comparison between MHN and MHS human subjects faces difficulties of a different nature. The differences at the cellular level that we and others have described result from post-translational modifications (phosphorylation, lysis) and have only been revealed by comparisons between groups. Together with the difficulties with recruitment (which we mentioned earlier in this response), these considerations justify a lower priority for the exploration of transcriptomics in human subjects.